# RNA degradation by the plant RNA exosome involves both phosphorolytic and hydrolytic activities

Natalia Sikorska [1], Hélène Zuber [1], Anthony Gobert [1], Heike Lange [1] & Dominique Gagliardi [1]

The RNA exosome provides eukaryotic cells with an essential 3′–5′ exoribonucleolytic activity, which processes or eliminates many classes of RNAs. Its nine-subunit core (Exo9) is structurally related to prokaryotic phosphorolytic exoribonucleases. Yet, yeast and animal Exo9s have lost the primordial phosphorolytic capacity and rely instead on associated hydrolytic ribonucleases for catalytic activity. Here, we demonstrate that *Arabidopsis* Exo9 has retained a distributive phosphorolytic activity, which contributes to rRNA maturation processes, the hallmark of exosome function. High-density mapping of 3′ extremities of rRNA maturation intermediates reveals the intricate interplay between three exoribonucleolytic activities coordinated by the plant exosome. Interestingly, the analysis of RRP41 protein diversity across eukaryotes suggests that Exo9's intrinsic activity operates throughout the green lineage, and possibly in some earlier-branching non-plant eukaryotes. Our results reveal a remarkable evolutionary variation of this essential RNA degradation machine in eukaryotes.

[1] IBMP, CNRS, University of Strasbourg, 12 rue du général Zimmer, 67000 Strasbourg, France. Correspondence and requests for materials should be addressed to D.G. (email: dominique.gagliardi@ibmp-cnrs.unistra.fr)

The RNA exosome is a multi-subunit complex, which is essential for the 3′–5′ processing and turnover of many RNA substrates including rRNA precursors and maturation by-products[1–5]. The core of the eukaryotic exosome (Exo9) consists of nine subunits: six RNase PH-like proteins (Rrp41, Rrp42, Rrp43, Rrp45, Rrp46, and Mtr3) and three RNA-binding proteins (Rrp4, Rrp40, and Csl4). Exo9 adopts a barrel-like structure, which is reminiscent of the structure of bacterial polynucleotide phosphorylases (PNPases) and archaeal exosomes[1–3,5,6]. These prokaryotic exoribonucleases are processive phosphorolytic enzymes with three catalytic sites inside the central channel of the barrel. By contrast, yeast and human Exo9s are catalytically inert due to mutations of amino acids critical for phosphorolysis. Therefore, the catalytic activity of the yeast and human RNA exosome depends strictly on the association of Exo9 with the hydrolytic ribonucleases Rrp6 and Rrp44/Dis3, or the paralog Dis3L in the cytosol of human cells[4,7–10].

The subunit composition of Exo9 and its interaction with a RRP44-like hydrolytic ribonuclease is conserved in Arabidopsis[11,12]. By contrast, there is no report yet demonstrating a physical association of Exo9 with any of the three RRP6-like exoribonucleases (RRP6L1-3) encoded by the Arabidopsis genome[11–13]. Of note, RRP6L1 and RRP6L2 have both independent and overlapping roles with the exosome. For instance, RRP6L2 shares common rRNA maturation substrates with the exosome, but it also processes a specific 18S rRNA intermediate, which is not a substrate of the exosome[13,14]. In addition, both RRP6L1 and RRP6L2 are involved in transcriptional gene silencing either independently of the exosome or with overlapping roles[15–17]. Finally, both RRP6L1 and RRP6L2 regulate the levels of transcripts antisense to FLOWERING LOCUS C (FLC) and this action does not require the exosome[18]. The role of RRP6L3 is entirely unknown and its functional relationship with the exosome remains elusive. It is also unknown whether plant Exo9 has retained an intrinsic ribonucleolytic activity similar to its structurally related complexes in prokaryotes or whether plant Exo9 is inactive as its yeast and animal counterparts. To date, three individual plant Exo9 subunits, RRP4, RRP41, and RRP46, were proposed to possess catalytic activity based on assays using recombinant proteins[19–21]. According to today's knowledge of the RNA exosome's atomic structure and of key residues required for ribonucleolytic activity, RRP41 remains the best candidate subunit for conferring activity to plant Exo9. Yet, RRP41's phosphorolytic activity remains to be formally demonstrated using catalytic mutants. More importantly, whether RRP41 is active upon exosome complex assembly and whether this activity performs any in vivo function are fully open questions.

Here, we demonstrate that the purified Arabidopsis exosome core complex is endowed with a phosphorolytic activity conferred by the RRP41 subunit. In vivo, this intrinsic phosphorolytic activity of Exo9 cooperates with the hydrolytic exoribonucleolytic activities of RRP6L2 and RRP44 for the elimination of rRNA maturation by-products and the processing/degradation of 5.8S rRNA precursors, two prototypical functions of the RNA exosome in eukaryotes. While the exoribonucleolytic activity of RRP44 is processive, Exo9's activity is distributive and both in vitro and in vivo data indicate that it can trim RNA substrates. Altogether, our results demonstrate that a eukaryotic core exosome can be catalytically active. Interestingly, the amino acids critical for RRP41's activity are remarkably conserved deep into the green lineage, suggesting that an active phosphorolytic Exo9 operates in most plant species. Moreover, several Amoebozoa, the human pathogen Naegleria fowleri or Capsaspora owczarzaki, a single-celled eukaryote, which is a close unicellular relative of metazoans, also possess a potentially active RRP41, raising the interesting possibility that an active Exo9 might exist in non-plant eukaryotes.

## Results

**Conservation of amino acids required for phosphorolysis.** The phosphorolytic activity of exoribonucleases, such as bacterial PNPases and the archaeal exosome depends on amino acids involved in the binding of the RNA substrate and in the coordination of inorganic phosphate (Pi) and divalent metal ions. Although RRP41 subunits of eukaryotic Exo9s show the highest conservation of amino acids necessary for activity, yeast and animal RRP41 subunits lack amino acids crucial for Pi coordination[8]. Yet, some amino acids required for phosphorolytic activity were noticed to be present in RRP41 from Arabidopsis and rice[8]. Nowadays, a comprehensive structural view of the phosphorolytic mechanism is proposed from several crystal structures of bacterial PNPases and archaeal exosomes, some of them in complex with Pi and RNA[22–30]. In the archaeal exosome, amino acids critical for catalysis belong to Rrp41 but are located near the interface of Rrp41–Rrp42 dimers. In addition, amino acids from both subunits bind the four nucleotides at the 3′ end of the RNA substrate. Together, these observations provide a rational explanation as to why Rrp41 alone is not active in vitro, whereas a Rrp41-Rrp42 dimer is[23,25,31,32]. To test for the conservation in Arabidopsis of all the amino acids required for phosphorolytic activity, we compared RRP41 sequences from Arabidopsis thaliana, Sulfolobus solfataricus, Archaeoglobus fulgidus, Pyrococcus abyssi and Methanothermobacter thermautotrophicus. We also aligned RRP42 sequences from these Archaea with Arabidopsis RRP45A and RRP45B. Both RRP45A and RRP45B isoforms are incorporated into Exo9 in Arabidopsis[12] and the RRP41-RRP45A/B dimer represents one of the three structural counterparts of the RRP41-RRP42 dimers of archaeal exosomes. Both alignments show that all the amino acids involved in RNA binding and in the coordination of divalent metal ions and Pi required for the phosphorolytic activity of archaeal exosomes are conserved in Arabidopsis (Fig. 1a and b). The structural modeling of the RRP41-RRP45A dimer also suggests the existence of functional Pi and $Mg^{2+}$ coordination sites in Arabidopsis (Fig. 1c). Therefore, Exo9 in Arabidopsis may have retained an intrinsic phosphorolytic activity unlike its inactive yeast and animal counterparts.

**Arabidopsis Exo9 has a phosphorolytic catalytic activity.** To test this hypothesis, we affinity purified Exo9 from rrp41 null mutant Arabidopsis plants expressing either wild type (RRP41$^{WT}$) or catalytic mutant RRP41 proteins. We choose mutations that target either only the Pi coordination site (RRP41$^{Pi-}$), or both the Pi and $Mg^{2+}$ coordination sites in the catalytic pocket (RRP41$^{Pi-Cat-}$) (Fig. 2a). Wild type and mutated RRP41 proteins are C-terminally fused to myc or GFP and all transgenes are expressed under the control of the endogenous RRP41 promoter. Because the rrp41 null mutation is lethal, plants heterozygous for the rrp41 null allele were transformed, selfed, and plants expressing the wild type or mutated RRP41 transgene in the homozygous rrp41 null background were obtained from the progeny. Wild type or mutated versions of RRP41-GFP fusion proteins displayed identical cytosolic and nuclear localization patterns: they were enriched in nucleoli and present at similar levels in nucleoplasm and cytoplasm as illustrated in root cells (Supplementary Fig. 1a). An identical intracellular distribution was observed in a line expressing another Exo9 subunit, RRP4, fused to GFP (Supplementary Fig. 1a). Although the lack of a suitable antibody prevented us to compare the expression between the tagged and endogenous RRP41s, we verified that all fusion proteins were expressed at similar levels in selected lines (Supplementary Fig. 1b). Importantly, gel filtration analyses indicated that neither wild type nor mutated RRP41 proteins are detected as monomeric proteins but

rather as part of a high molecular weight complex (Supplementary Fig. 1c). To determine whether this high molecular weight complex corresponds to the RNA exosome, myc affinity-purified fractions were submitted to mass spectrometric analysis. The identification of all expected Exo9 subunits demonstrated the incorporation of RRP41$^{WT}$, RRP41$^{Pi-}$ or RRP41$^{Pi-Cat-}$ into Exo9 (Supplementary Fig. 1d). Altogether, these data indicate that both wild type and mutated RRP41 versions behave indifferently in terms of expression, localization, and integration into Exo9.

To test for Exo9's phosphorolytic activity, purified Exo9s containing RRP41$^{WT}$, RRP41$^{Pi-}$, or RRP41$^{Pi-Cat-}$ subunits were incubated with a 5′-radiolabeled $(U)_{21}$ RNA substrate in the presence or absence of inorganic phosphate (Fig. 2b). Phosphate-stimulated degradation of the RNA substrate was reproducibly observed with Exo9-RRP41$^{WT}$, but never with the catalytic mutants (Fig. 2b and Supplementary Figs. 2 and 3). Next, we used thin layer chromatography (TLC) analysis to show that degradation of a 3′-labeled RNA substrate by Exo9-RRP41$^{WT}$ releases nucleoside diphosphates (Fig. 2c). Finally, we demonstrate that the reaction is reversible when nucleoside diphosphates are present in excess, i.e. plant Exo9 can synthesize RNA tails (Fig. 2d). To exclude the possibility that the observed in vitro activity is due to RRP41 not incorporated into Exo9, we compared the phosphorolytic activity of Exo9s immunopurified from lines expressing either GFP-tagged RRP4 or GFP-tagged RRP41. Both immunopurified fractions have identical degradation and synthesis activities, confirming that the observed activity is due to RRP41 incorporated into Exo9 (Supplementary Fig. 3).

An interesting and major distinction from bacterial PNPases and the archaeal exosome, which are processive phosphorylases, is the distributive nature of Exo9's activity. This distributivity is evidenced by the progressive degradation of the RNA substrate, while both full-length substrates and end-products are visible in the case of the processive bacterial PNPase (Fig. 2e). Similar results were obtained using a RNA substrate of distinct nucleotide composition (Supplementary Fig. 2). Altogether, our results prove that *Arabidopsis* Exo9 has an intrinsic distributive phosphorolytic activity provided exclusively by the RRP41 subunit.

**Exo9 activity acts on specific rRNA maturation by-products.**
To reveal evidence of Exo9's intrinsic activity in vivo, we analyzed the degradation or trimming of archetypical substrates of the exosome during rRNA maturation: the 5′ external transcribed spacer (5′ ETS) and 5.8S rRNA precursors. Similar to other eukaryotes, *Arabidopsis* mature rRNAs are produced from a large polycistronic precursor transcript, in which the sequences of mature 18S, 5.8S and 25S rRNAs are separated by internal transcribed spacers (ITS1 and ITS2) and flanked by external transcribed spacers (5′ and 3′ ETS, respectively) (see Supplementary Fig. 4 for a diagram). In *Arabidopsis*, two endonucleolytic cleavages at sites P and P′ excise a 481 nt fragment from the 1.8 kb 5′ ETS[13,33] (Fig. 3a and Supplementary Fig. 4). Degradation of the P-P′ fragment requires the exosome as its elimination is impaired by disrupting Exo9 in lines silenced for RRP41 or RRP4 expression, or by downregulating RRP44, MTR4, and RRP6L2[11,13,14,34,35]. The 3′–5′ degradation of P-P′ produces

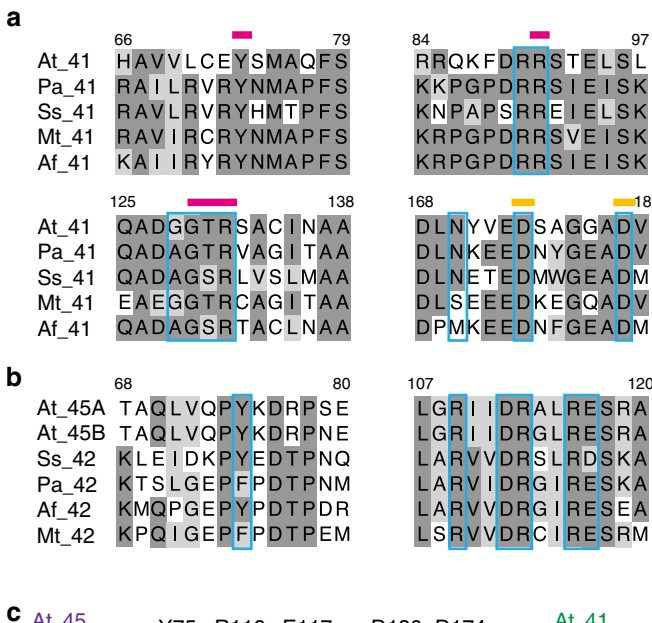

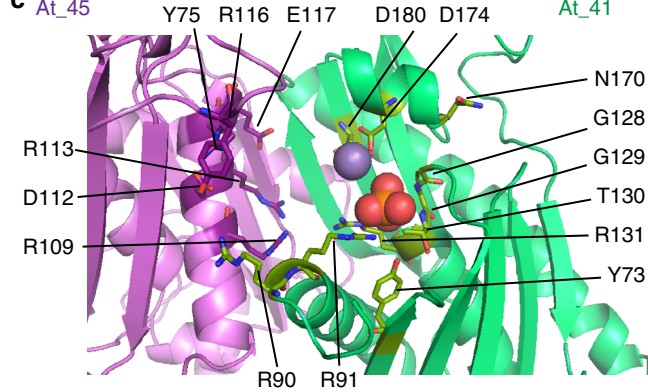

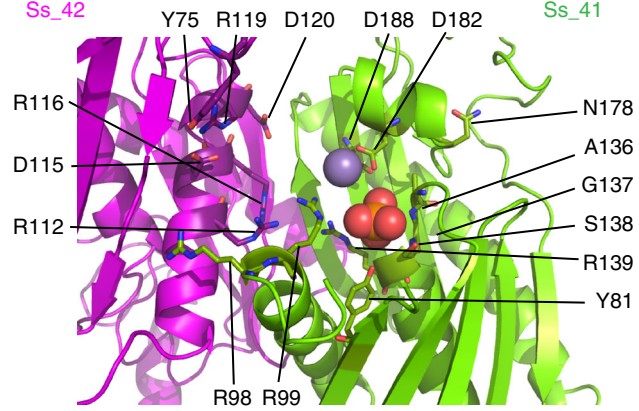

**Fig. 1** Amino acids crucial for the phosphorolytic activity of archaeal exosomes are conserved in *Arabidopsis* RRP41-RRP45 subunits. The *Arabidopsis* RRP41, RRP45A, and RRP45B sequences were aligned to their respective RRP41 and RRP42 counterparts from four archaeal species for which exosome structure is known: Ss, *Sulfolobus solfataricus*; Pa, *Pyrococcus abyssi*; Af, *Archaeoglobus fulgidus*; Mt, *Methanothermobacter thermautotrophicus*. Amino acid numbering refers to the *Arabidopsis* sequences. Amino acids involved in RNA binding are boxed in blue, amino acids involved in Pi and Mg$^{2+}$ coordination are marked with magenta and yellow bars, respectively. **a** Partial alignments of *Arabidopsis* and four archaeal RRP41 sequences showing the conservation of amino acids involved in RNA binding, Pi and Mg$^{2+}$ coordination. **b** Partial alignments of *Arabidopsis* RRP45A and RRP45B with four archaeal RRP42 sequences showing the conservation of relevant amino acids involved in RNA binding. **c** Zoom in the catalytic pocket of the modeled At_RRP41/45 dimer (upper panel) and of the Ss_RRP41/42 dimer (lower panel). Mg$^{2+}$ and Pi are shown as violet and red balls, respectively. Amino acids whose side chains are involved in RNA binding, Pi and Mg$^{2+}$ coordination are labeled

several P-P1 degradation intermediates[14,34,36]. Four P-P1 degradation intermediates are detected by northern blot analysis in untransformed Col-0 and *rrp41* RRP41[WT] (Fig. 3b, lanes 1–2). Strikingly, two of these P-P1 fragments are missing in *rrp41* lines expressing RRP41 catalytic mutants (Fig. 3b, lanes 3–4). This alteration in the pattern of P-P1 intermediates reveals that Exo9 is

active in vivo and indicates that its activity participates in the elimination of the P-P′ rRNA maturation by-product.

To further characterize P-P1 intermediates, their 3′ extremities were mapped at high density using 3′RACE-seq, a high-throughput sequencing-based strategy (Supplementary Table 1). In line with the northern analysis, four major P-P1 fragments are detected in Col-0 plants and *rrp41* RRP41[WT] (Fig. 3c). These P-P1 fragments are thereafter named P161, P168, P176, and P186 to reflect their respective length in nucleotides (+1 is defined as the first nucleotide of the P-P′ fragment). The 3′RACE-seq profiles for P-P1 intermediates are identical between Col-0 and *rrp41* RRP41[WT] samples, indicative of a full complementation by the transgene. In agreement with the northern data, P161 and P176 intermediates are absent in *rrp41* RRP41[Pi-] and *rrp41* RRP41[Pi-Cat-] samples (Fig. 3c and Supplementary Fig. 4b). Therefore, P161 and P176 are exclusively produced by Exo9's intrinsic activity, possibly by nibbling P168 and P186. Altogether, these data demonstrate that specific rRNA maturation by-products that are generated during the elimination of the 5′ ETS are substrates of the intrinsic activity of Exo9 in *Arabidopsis*.

**Extensive exoribonucleolytic interplay during 5′ ETS decay.** Next, we determined the respective effects of Exo9's activity and of the other two exoribonucleases associated with exosome function, RRP6L2 and RRP44, on 5′ ETS degradation. As the null *rrp44* mutation is lethal[37], we used an established amiRNA[35] to test the effect of RRP44 downregulation in the *rrp6L2* background. The accumulation of P-P′, a known substrate of RRP44, was used as a functional indicator of RRP44 downregulation (Fig. 3b, lanes 8–10). A slight accumulation of P176 was detected by 3′RACE-seq in *rrp6L2* and *rrp6L2 rrp41* RRP41[WT] as compared with Col-0 and *rrp41* RRP41[WT], respectively (Fig. 3d). The most straightforward interpretation of this observation is that P176 is a substrate of RRP6L2, albeit an indirect effect of the lack of RRP6L2 on RRP44's activity cannot be excluded. Similar to what we observed before, loss of Exo9's activity in *rrp6L2 rrp41* RRP41[Pi-Cat-] plants resulted in the absence of P161 and P176, confirming that these intermediates are generated by Exo9 (Fig. 3b, lane 7 and Fig. 3d).

Downregulation of RRP44 in the *rrp6L2* background resulted in the further accumulation of P176 in *RRP44KD rrp6L2* (Fig. 3b, lanes 8–10 and Fig. 3e) as compared to *rrp6L2* single mutants (Fig. 3b, lanes 5–7 and Fig. 3d). This accumulation indicates that

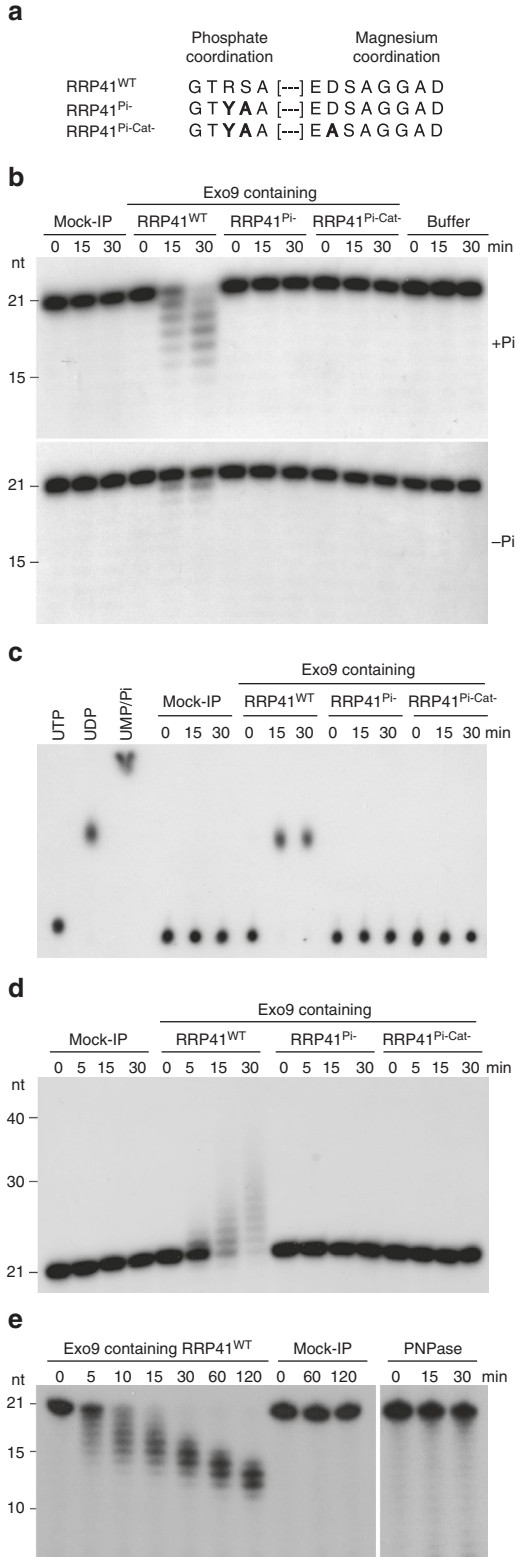

**Fig. 2** *Arabidopsis* Exo9 has a distributive phosphorolytic activity conferred by the RRP41 subunit. **a** Mutations introduced to inactivate RRP41. Mutated amino acids are shown in bold. The mutated Pi coordination site and surrounding amino acids in RRP41[Pi-] and RRP41[Pi-Cat-] mimics the GGTYAA sequence found in the inactive human RRP41. **b** Exo9's activity is due to RRP41 and is stimulated by Pi. A 5′ [32P]-labeled oligo(U)21 RNA substrate was incubated for indicated times with a mock-IP fraction or immunopurified Exo9 complexes containing either RRP41[WT], RRP41[Pi-], or RRP41[Pi-Cat-]. Reaction products were run on a denaturing acrylamide gel and analyzed by autoradiography. +Pi and −Pi indicate the presence of 3.5 mM Pi and the absence of exogenously added Pi, respectively. **c** Exo9's degradation activity produces nucleoside diphosphates. A 3′ [32P]-labeled oligo(U) RNA substrate was incubated with a mock-IP fraction or immunopurified Exo9 complexes containing either RRP41[WT], RRP41[Pi-], or RRP41[Pi-Cat-]. Reaction products were analyzed by thin layer chromatography to separate nucleoside mono-phosphate, di-phosphate, and tri-phosphate. **d** Exo9 can synthesize RNA tails. In vitro assays were performed as in **a** except that Pi was replaced by 1 mM UDP. **e** Exo9 has an intrinsic distributive activity. Degradation assays in the presence of 3.5 mM Pi were as in **a** but over a period of 120 min as indicated. Bacterial PNPase was used as a positive control for a processive phosphorylase activity

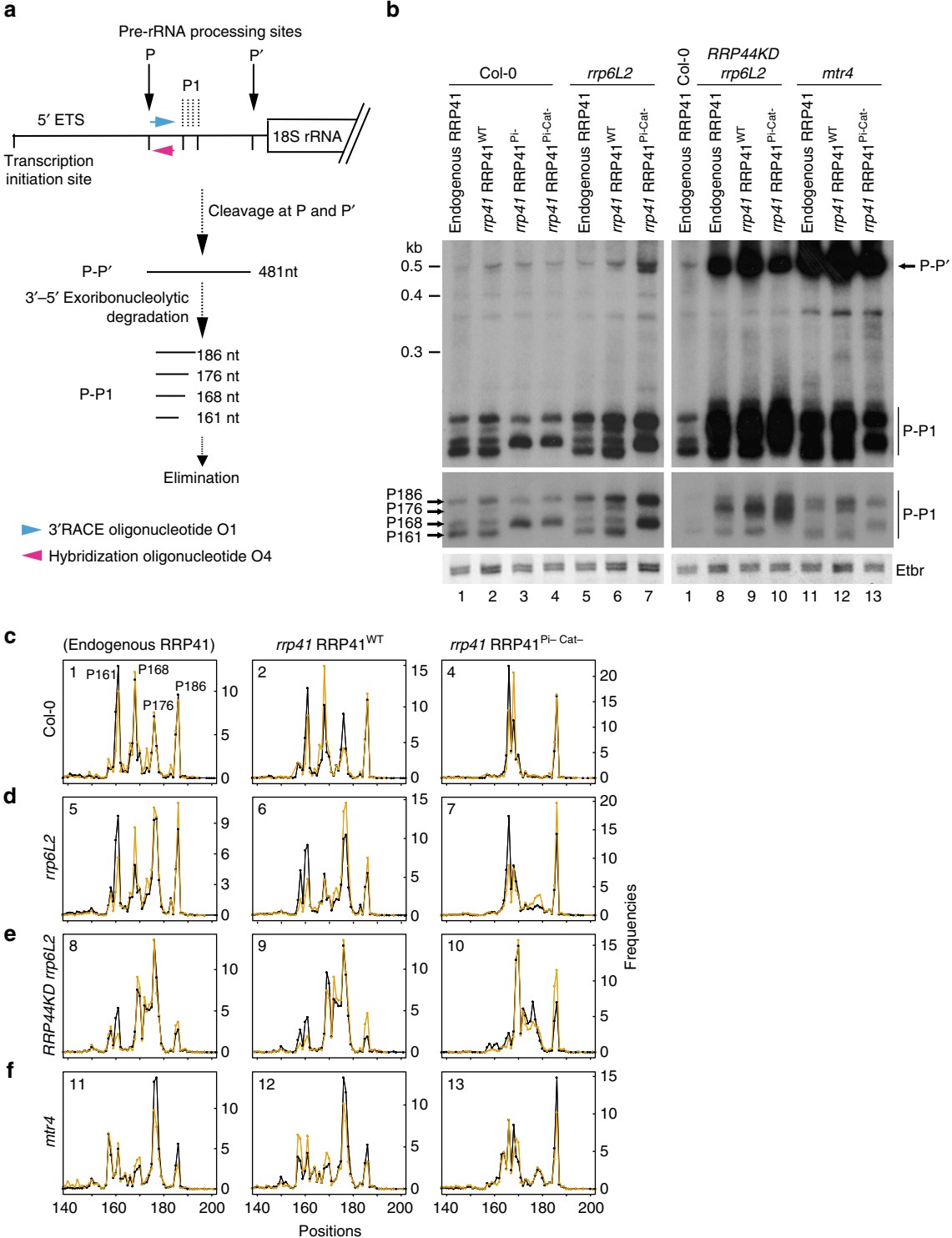

**Fig. 3** Exo9 activity participates in the elimination of rRNA maturation by-products. **a** Diagram of the 5′ ETS and its degradation intermediates. A portion of the 35S rRNA precursor comprising the 5′ ETS is drawn at the top. Numbers refer to the transcription start site. Vertical arrows above the diagram indicate processing sites. Key steps leading to the elimination of the 5′ ETS are sketched below. The location of the hybridization oligonucleotide O4 used for Northern analysis is shown by a magenta arrow. The location of the oligonucleotide O1 used for 3′RACE-seq is shown by a blue arrow. **b** Northern blot analysis of Exo9's specific in vivo substrates during the elimination of the 5′ ETS. The main 5′ ETS degradation intermediates are noted as P-P′ and P-P1 on the right. Two exposures of the lower part of the blot are shown. A portion of the Ethidium bromide (Etbr)-stained gel is used as control for loading. Lane numbers at the bottom refer to the panels shown in **c–f**. **c–f** High density mapping of 3′ extremities of P-P1 degradation intermediates by 3′RACE-Seq. Density profiles are shown for positions +140 to +200, +1 corresponding to the first nucleotide of the P-P′ fragment. Two biological replicates are shown in black and orange, respectively. The presence of endogenous RRP41 or the complementation of the *rrp41* mutations by RRP41^WT or RRP41^Pi-Cat-, respectively, is indicated at the top of the panels. The Col-0, *rrp6L2*, *RRP44KD rrp6L2* or *mtr4* genetic backgrounds are indicated on the left in **c**, **d**, **e**, and **f**, respectively. Numbering of each panel refers to the lane numbers of the northern blot shown in **b**

both RRP6 and RRP44 are required for the efficient degradation of P176. As expected, inactivating RRP41's activity in the *RRP44KD rrp6L2* background had a severe impact on the accumulation of P176. Yet, this intermediate is still detectable in both replicates of *RRP44KD rrp6L2 rrp41* RRP41$^{Pi-Cat-}$. This observation, together with the occurrence of a new major P169/170 intermediate instead of P168 (Fig. 3e), suggest that, when all three catalytic activities associated with the plant exosome are impaired, additional yet unknown ribonucleolytic activities can act on P-P1 intermediates.

The role of Exo9 in generating specific intermediates including P176 is further evidenced in *mtr4* mutants. In *Arabidopsis*, the RNA helicase MTR4 is crucial for the nucleolar functions of the exosome, while its paralogue HEN2 assists the exosome in the nucleoplasm[12]. In the absence of MTR4, P-P' and P-P1 intermediates accumulate markedly when compared to Col-0 and *rrp41* RRP41$^{WT}$ plants (Fig. 3b, lanes 11–13). The accumulation of P176 reflects the expected role of MTR4 in supporting RRP6L2 and RRP44 activities, both of which play a role in P176 elimination. In addition, the 3′RACE-seq profiles reveal a continuous nibbling of P-P1 intermediates from +155 to +170 in *mtr4* and *mtr4 rrp41* RRP41$^{WT}$ (Fig. 3f). Inactivating Exo9's activity had two main effects in *mtr4* background: it averts the accumulation of P176, and it prevents the accumulation of P-P1 species smaller than 161 nt, another evidence that Exo9's activity can nibble P-P1 intermediates (Fig. 3f).

Altogether, our results show that Exo9's activity cooperates with RRP6L2 and RRP44 to eliminate the 5′ ETS in *Arabidopsis*. Exo9 is able to nibble 5′ ETS degradation intermediates by 5–10 nt and its activity is not fully redundant with that of the known catalytic subunits of the exosome. Exo9's action either provides intermediates that are further processed by RRP6L2 or RRP44 (e.g. P176), or is able to nibble intermediates further than RRP6L2 and RRP44 (e.g. P161 or smaller species in the absence of MTR4).

**Indirect impact of Exo9's activity on P-P1 tailing**. Up to 35% of the P-P′ degradation intermediates analyzed by 3′RACE-seq have 3′ nucleotides that do not match the reference sequence (Fig. 4a). Many of these mismatches correspond to a single nucleotide (Fig. 4a) and it cannot be excluded that a proportion of them is due to sequence polymorphisms rather than to untemplated nucleotide additions. However, bona fide nucleotide extensions are definitely detected and are mostly composed of adenosines (from 74 to 94% As for tails from 2 to 30 nt). Loss of the ribonucleases RRP6L2 or RRP44, or of the RNA helicase MTR4, increases their proportion as compared with Col-0 (Fig. 4a). Unexpectedly, the sole inactivation of Exo9's activity in Col-0 background results in a decrease in the proportion of tailed P-P′ degradation intermediates (Fig. 4a). This decrease does not reflect the ability of Exo9 to synthesize tails or the enhanced degradation of tailed P-P1 intermediates when Exo9 is inactivated. Instead, the general diminution in tailing is due to a strict discrimination between P-P1 intermediates' ability to be tailed. Indeed, out of the four main P-P1 intermediates present in Col-0 and *rrp41* RRP41$^{WT}$, only P161 and P176 are modified by the 3′ addition of untemplated nucleotides (Fig. 4b). The accumulation of tailed P-P1 intermediates is almost abolished when RRP41 is inactive (Fig. 4b) likely because the P161 and P176 species are not generated and only those intermediates are modified by nucleotide addition. Therefore, Exo9's activity has an indirect impact on the frequency of P-P1 intermediates tailed by nucleotide addition. However, the tails still observed in the absence of active Exo9 are longer (Fig. 4c). Because tail length of RNA substrates destined for degradation is usually increased in exoribonuclease mutants, this is another indication that Exo9's activity directly participates in the degradation of P-P1 intermediates.

**Exo9 contributes to 5.8S rRNA metabolism**. Next, we performed 3′RACE-seq experiments to analyze 3′ processing of the 5.8S rRNA, another classical function of the RNA exosome in eukaryotes (Supplementary Table 2). First, we mapped the mature 3′ extremity of 5.8S rRNA in Col-0 by using a forward primer within the 5.8S sequence (Supplementary Fig. 5). Most reads (98.6%) mapped 1 nt upstream of the currently annotated 3′ end of 5.8S rRNA (Supplementary Fig. 5). A similar experiment in the *rrp6L2* mutant detected the same 3′ terminal nucleotide for mature 5.8S rRNA, and that the shortest 5.8S rRNA precursor is extended by 11/12 nt (Supplementary Fig. 5). To favor the amplification of precursors rather than mature 5.8S rRNA, we used a primer ending 4 nt downstream of the 5.8S rRNA mature 3′ end for the high density mapping of 3′ extremities of 5.8S precursors across genotypes (Fig. 5a and b). This experiment confirmed the presence of the +11/12 species upon inactivation of RRP6L2, MTR4 or downregulation of RRP44 (Fig. 5a). Notably, the 5.8S + 11/12 intermediate still accumulates in *RRP44KD rrp6L2 rrp41* RRP41$^{Pi-Cat-}$, strongly suggesting that this species is likely produced by an alternative ribonucleolytic activity that is not associated with the RNA exosome. Therefore, we propose that its accumulation reveals the impairment of exosome function rather than being the bona fide substrate of Exo9, RRP6L2, or RRP44. By contrast, 5.8S precursors extended by 17–24 nt accumulate slightly more in the absence of RRP41's activity (Fig. 5a). Interestingly, those intermediates are detected when either RRP6L2 or both RRP6L2 and RRP44 are impaired, suggesting that Exo9's activity could compete for substrates with the classical ribonucleolytic activities linked to exosome function. Because of the few qualitative differences in the 5.8S rRNA precursor profiles detected by 3′RACE-seq, the intrinsic activity of the core complex Exo9 could be mostly redundant with the hydrolytic activities linked to exosome function. This hypothesis was confirmed by comparing the accumulation of 5.8S rRNA precursors in plants possessing or lacking Exo9's activity by northern blot analysis (Fig. 5c). This analysis confirmed that the sole inactivation of RRP41's activity has no major effect on the accumulation of 5.8S precursors (Fig. 5c, lanes 1–4). By contrast, a marked accumulation of the smallest 5.8S rRNA precursor (i.e. 5.8S + 11/12 nt) is observed when RRP41's activity is abrogated together with the inactivation of RRP6L2 or with the downregulation of RRP44 (Fig. 5c, lanes 5–10).

Altogether, these results prove that 5.8S rRNA precursors are endogenous RNA substrates of Exo9's intrinsic activity, and that the exoribonucleolytic activities of RRP44 and RRP6L2 together with Exo9 participate in the processing or degradation of 5.8S rRNA precursors in *Arabidopsis*.

**Lack of RRP41 activity impacts plant development and growth.** Plants expressing either active or inactive versions of RRP41 in the Col-0 wild type background do not show obvious growth phenotypes. By contrast, double mutants expressing inactivated RRP41 in *rrp6L2* or *mtr4* backgrounds, or triple mutants expressing inactivated RRP41 in the *rrp6L2 RRP44 KD* background have pronounced developmental defects as compared to their RRP41$^{WT}$-expressing counterparts (Fig. 6). Interestingly, the extent of the phenotypes observed for the mutant plants depends on the culture conditions. Short-day photoperiods slow down individual leaf growth, albeit total plant leaf area is increased because the transition to flowering is delayed[38]. Conditions that favor slow growth of individual leaves appear to minimize the impact of inactivated RRP41 on plant growth and development (Supplementary Fig. 6). Shifting plants to long day (16 h) photoperiods, which promotes faster growth of individual leaves, reveals that the combined inactivation of RRP41 together with the

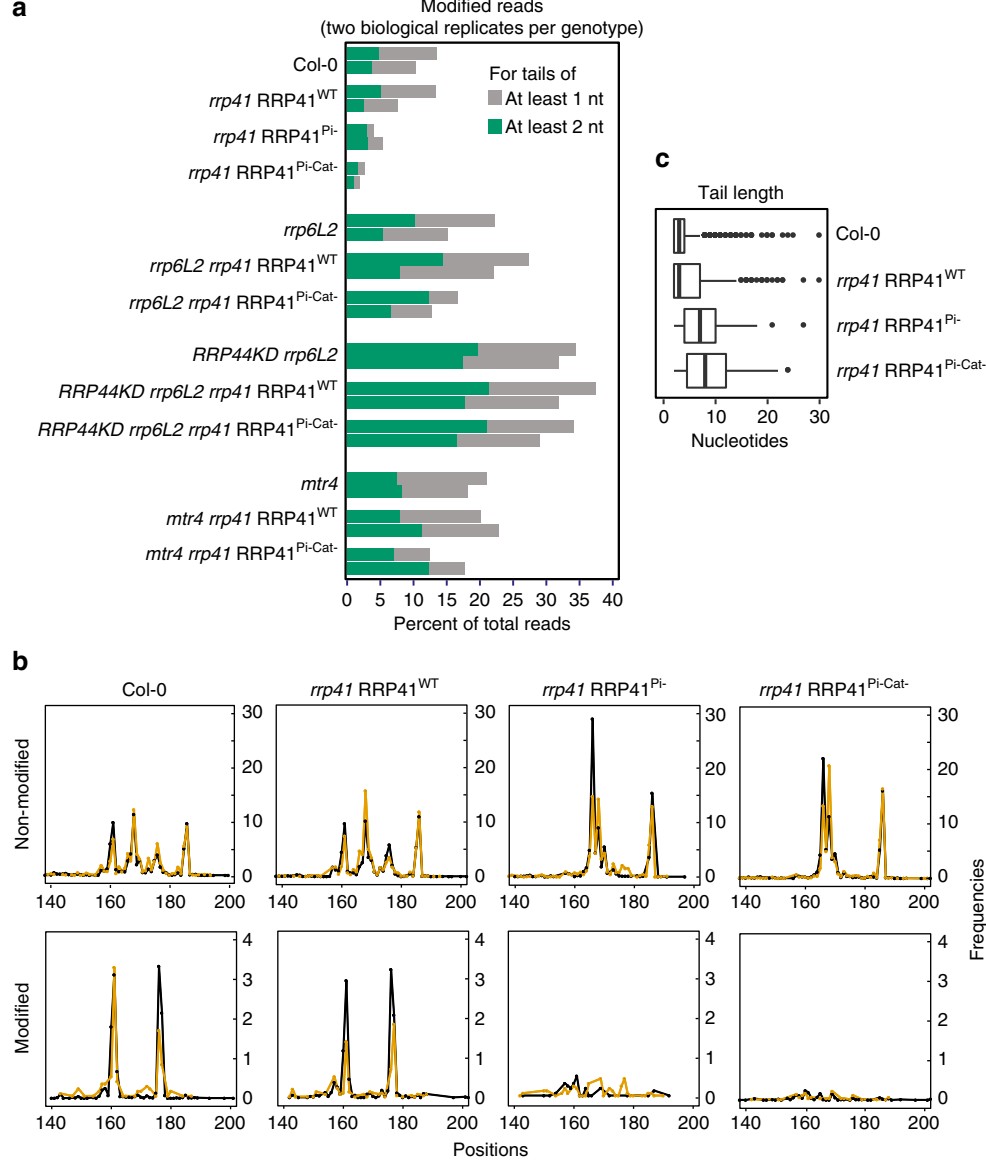

**Fig. 4** Impact of Exo9 activity on the tailing of 5′ ETS degradation intermediates. **a** Exo9 inactivation results in reduced RNA tailing. The proportion of reads corresponding to 5′ ETS degradation intermediates modified by nucleotide additions is shown in gray (from 1 to 30 nt) or green (from 2 to 30 nt) for two biological replicates. **b** Nucleotides are only added to specific P-P1 degradation intermediates. Reads corresponding to non-modified or modified P-P1 intermediates were split to generate the profiles shown as upper and lower panels, respectively. Two independent biological replicates are shown as black and orange lines. **c** Increase of tail length due to the inactivation of Exo9's activity. Box plot representation of tail lengths in nucleotides. The median is indicated by the black bars, the first and third quartiles by the box edges, the interquartile range by thin bars, the outliers by dots and the data range (except outliers) is shown by whiskers

*rrp6L2*, *rrp6L2*, and *RRP44 KD*, or *mtr4* mutations leads to a pronounced phenotype. In each of these mutant backgrounds, inactivating RRP41's activity severely impedes growth (Fig. 6a and Supplementary Fig. 6). In addition, lack of RRP41's activity in the *mtr4* background strongly affects leaf development as it leads to the frequent formation of cup-shaped leaves (Fig. 6a).

The phenotypic analysis of double and triple mutant plants supports the hypothesis that Exo9's activity participates in ribosome biogenesis. In particular, an aberrant venation pattern of cotyledons becomes evident upon RRP41's inactivation in *rrp6L2* mutants (Fig. 6b and Supplementary Fig. 7). An aberrant venation pattern was defined by the failure of the two early appearing veins to form a closed structure, which is usually observed in wild type cotyledons. This phenotype is due to impaired responsiveness to the phytohormone auxin and is

classically observed in case of defective ribosome biogenesis, either when rRNA maturation or the synthesis of ribosomal proteins is affected[39]. As aberrant vein patterns are already frequently observed in *mtr4* mutants we cannot measure an aggravating effect of inactivating RRP41 (Supplementary Fig. 7). By contrast, inactivation of Exo9's activity does significantly enhance the proportion of defective cotyledon vein patterns in the *rrp6L2* backgrounds (Fig. 6b and c). While less than 10% of *rrp6L2* or *rrp6L2* RRP41^WT cotyledons have disturbed vein patters, 68% of *rrp6L2* RRP41^Pi- Cat- plants display this characteristic phenotype ($p < 2.2e−16$). Hence, its occurrence is clearly linked to RRP41's inactivation and mirrors the severity of the molecular defects observed for 5.8 rRNA processing (Fig. 5c). Both the growth retardation under conditions that promote fast leaf growth and the impaired formation of vein pattern in

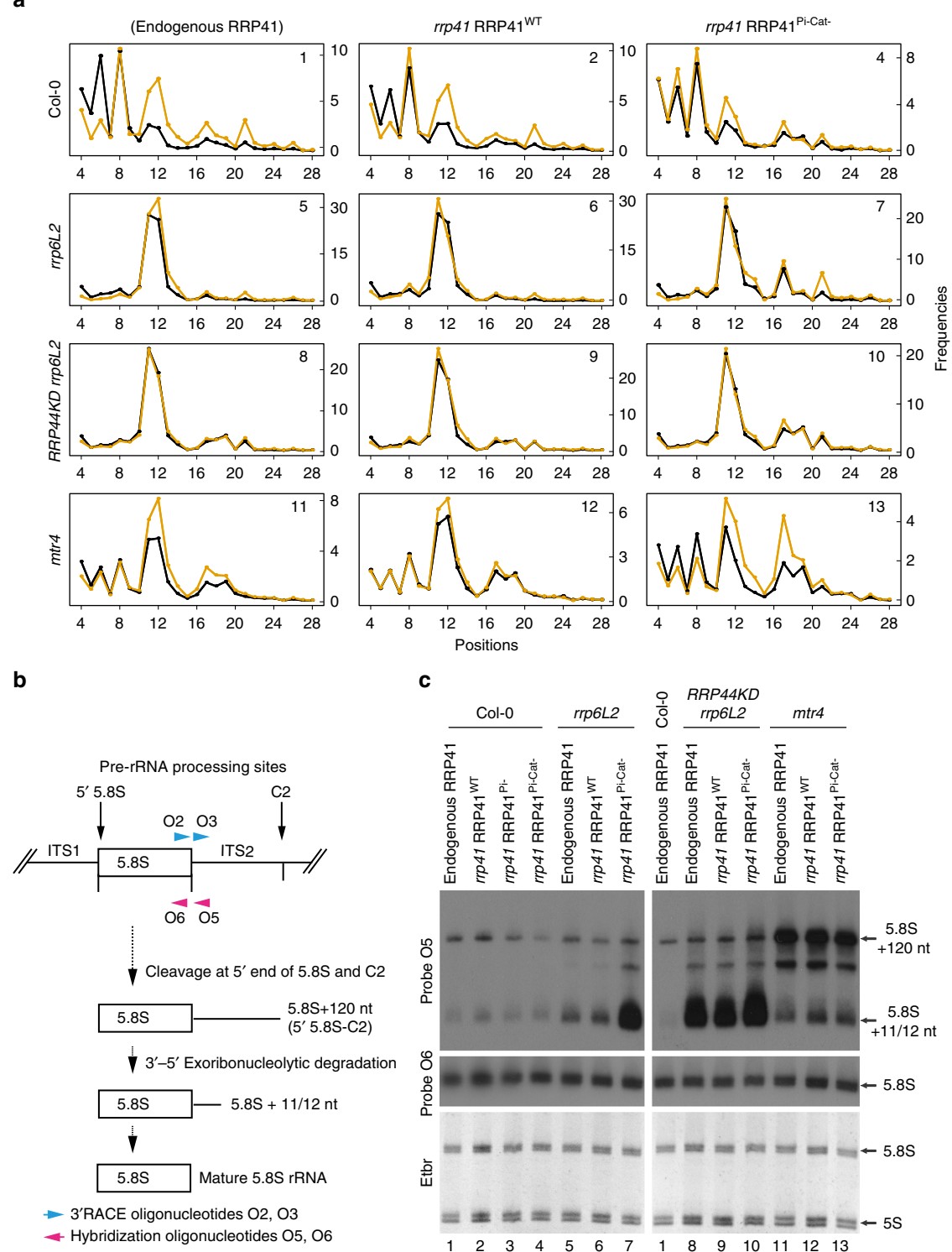

**Fig. 5** Exo9 activity is involved in processing or degradation of 5.8S rRNA precursors. **a** High density mapping of 5.8S rRNA precursors by 3′RACE-seq. Density profiles of 5.8S rRNA precursors degradation intermediates are shown for positions +4 to +28, +1 corresponding to the first nucleotide after the mature 5.8S rRNA (see Supplementary Fig. 5). Two replicates are shown in black and orange. The presence of the endogenous RRP41 or the complementation of the *rrp41* mutations by RRP41[WT] or RRP41[Pi-Cat-], respectively, is indicated at the top of the panels. The Col-0, *rrp6L2*, *RRP44KD rrp6L2*, or *mtr4* genetic background is indicated on the left. Numbering of each panel refers to the lane numbers of the northern blot shown in **c**. **b** Diagram of the 5.8S rRNA processing intermediates. Numbers refer to the transcription start site. Vertical arrows above the diagram indicate endonucleolytic processing sites. Key steps leading to the maturation of the 5.8S rRNA are sketched below. The location of the hybridization oligonucleotides O5 and O6 is shown by a magenta arrow. The location of the oligonucleotides used for 3′RACE-seq is shown by blue arrows. O2 was used to map mature 5.8S rRNA mature ends (Supplementary Fig. 5), O3 to map 3′ ends of 5.8S rRNA precursors in **a**. **c** Northern blot analysis of the accumulation of 5.8S rRNA precursors and mature 5.8S rRNA upon inactivation of Exo9's activity. A portion of the ethidium bromide (Etbr)-stained gel is used as loading control. Lane numbers refer to the panels in **a**

cotyledons can be linked to defective ribosome biogenesis and reveal the biological importance of Exo9's activity. Because single RRP41 mutations do not seem to affect plant growth under none of various culture conditions that we have tested, we propose that Exo9's activity could be important to confer robustness rather than specificity to exosome-mediated rRNA processing and degradation processes.

**Potential activity of RRP41 orthologues in eukaryotes.** The intrinsic phosphorolytic activity of Exo9 demonstrated here for

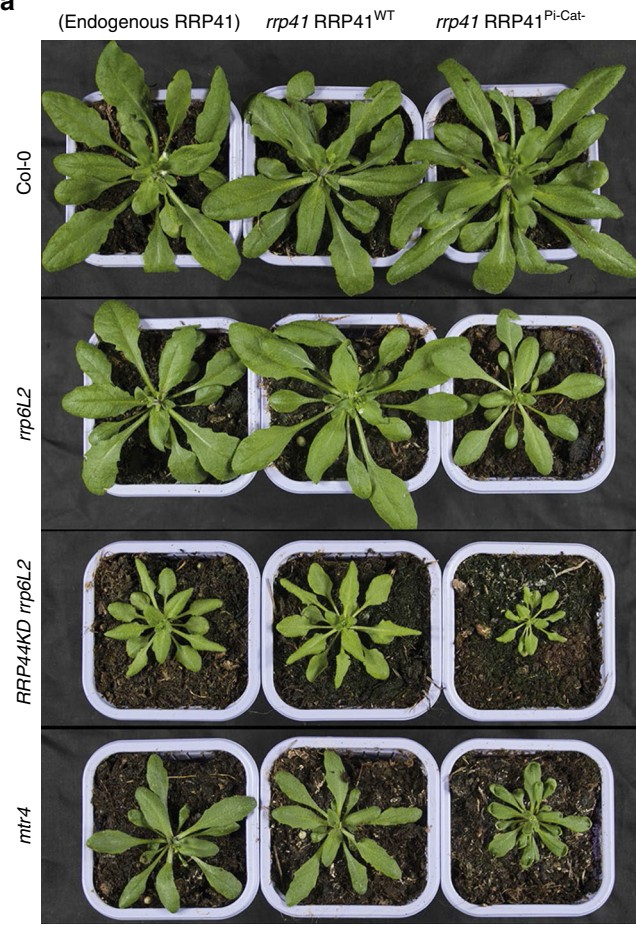

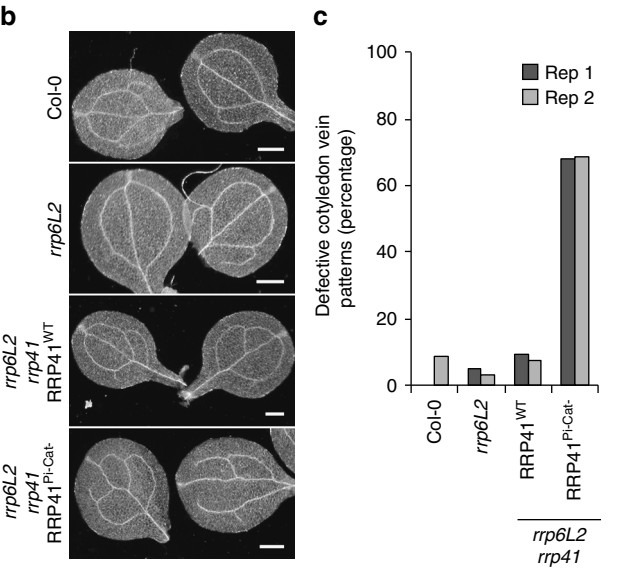

*Arabidopsis* is unlikely restricted to this single plant species. Indeed, analyzing RRP41 sequences from representative of different phylogenetic classes forming Archaeplastida revealed the conservation of amino acids critical for phosphorolytic active RRP41s in the plant lineage (Fig. 7 and Supplementary Data 1). Importantly, all but one genome of investigated Streptophyta seem to encode an active RRP41, from Streptophyte algae to Embryophytes (all land plants). Indeed, the amino acids responsible for phosphate coordination within the motif DGGTR (Fig. 7, magenta bars below sequence) are strictly conserved in 106 out of 107 genomes of land plants analyzed here. The only exception is for an Acrogymnosperma RRP41 protein containing the sequence DGGVK instead of DGGTR, and this alternative motif might still be able to coordinate Pi. The amino acid immediately following the canonical DGGTR motif is a S in most land plant species, except in mosses and worts. However, variation at this position is likely compatible with phosphorolytic activity because it is observed at the corresponding position of active RRP41s from archaeal exosomes (see Fig. 1). Besides Streptophyta, amino acids required for phosphorolytic activity are present in RRP41s in Chlorophyceae, in certain Prasinophyta and Trebouxiophyceae, and even in some red algae (Rhodophyta). Based on these observations, we propose that the phosphorolytic activity of Exo9 due to RRP41 is likely conserved in all land plants, as well as in representatives of green and red algae, illustrating a major difference between plant exosomes and their yeast and human counterparts.

Of note, potentially active RRP41 sequences are also present in some earlier-branching non-plant eukaryotes (Fig. 8 and Supplementary Data 2). Such species include several Amoebozoa, the human pathogen *N. fowleri* (Excavata) and the Opisthokonta *C. owczarzaki*, a single-celled eukaryote, which is a close unicellular relative of metazoans. These observations open the interesting possibility that a phosphorolytic active Exo9 might also operate in some non-plant eukaryotes.

## Discussion

The overall architecture of the exosome is remarkably conserved among eukaryotes. Moreover, all eukaryotic exosomes require a similar set of central co-factors involved in the recognition and degradation of its RNA substrates. Despite this high degree of conservation, we report a fundamental difference amongst eukaryotic exosomes: while the core exosome has lost its ribonucleolytic capacity in fungi and metazoans, the plant core exosome, as illustrated here for *Arabidopsis*, has retained a catalytic activity provided by the RRP41 subunit. Aminoacids required for RRP41's activity are indeed remarkably conserved in all land plants and most green algae. This suggests that the phosphorolytic activity of Exo9 is a salient feature of Chloroplastida and may even be present in some representatives of red algae. Our

**Fig. 6** Phenotype of Exo9 activity mutants. **a** Plants expressing endogenous RRP41^WT, RRP41^Pi-, or RRP41^Pi-Cat- in Col-0, *rrp6L2*, *RRP44KD rrp6L2*, or *mtr4* genetic backgrounds were grown for 6 weeks in conditions promoting fast growth. More pictures are presented in Supplementary Fig. 6. **b** Darkfield microscopy of cotyledons prepared from 7-day old seedlings of Col-0 and *rrp6L2* expressing either active or inactive versions of RRP41 as indicated on the left. Scalebars are 0.5 mm. Vein patterns are defective when the two early appearing veins fail to form a closed structure that is usually observed in wild type cotyledons. **c** Histogram showing the proportion of defective cotyledon vein patterns in Col-0, *rrp6L2*, *rrp6L2 rrp41* RRP41^WT, and *rrp6L2 rrp41* RRP41^Pi-Cat- for two biological replicates. The corresponding numbers and more pictures are provided in Supplementary Fig. 7

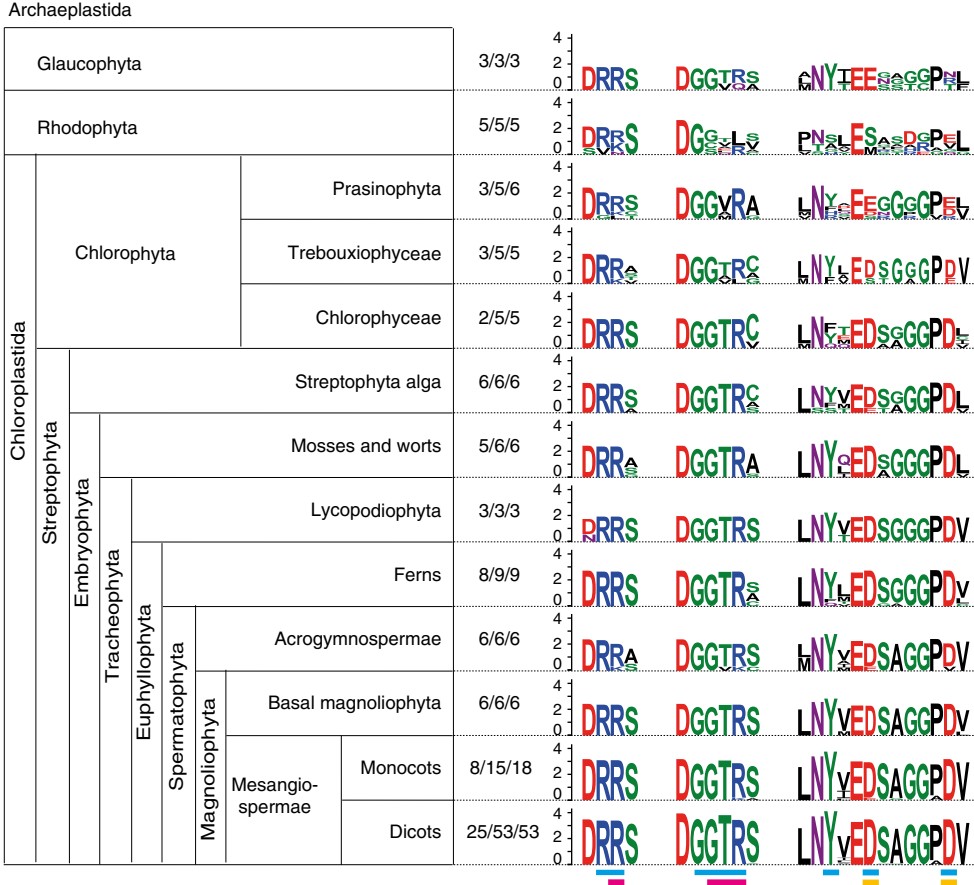

**Fig. 7** Conservation across the green lineage of amino acids critical for RRP41 activity. Sequence logos expressed in bits were calculated from sequence alignments of RRP41 orthologues from all divisions of Archaeplastida. The number of order/families/sequences used for each alignment is indicated. The bars below the sequence logos indicate amino acids involved in RNA binding (blue), Pi coordination (magenta) and Mg$^{2+}$ coordination (yellow)

current view that eukaryotic core exosomes are inactive must therefore be reconsidered to include the notion that plant Exo9s are active. Besides plants, an active RRP41 subunit might also be present in a few representatives of the other major eukaryotic groups.

We show here that the *Arabidopsis* core exosome contains a single active site. Therefore, it also stands out as compared to its prokaryotic counterparts, the bacterial PNPase and archaeal exosome, which have three active sites in their central channel. Could this difference account for the intrinsic distributive activity of the plant Exo9 while bacterial PNPases and archaeal exosomes are processive enzymes? In fact, the processivity of these prokaryotic exoribonucleases is primarily determined by the RNA binding affinity at the entrance of the channel[6,40,41]. Mutations that decrease RNA binding affinity at the channel entrance are sufficient to switch a processive *S. solfataricus* exosome into a distributive enzyme[41]. Albeit all three active sites inside the channel contribute to degradation, the RNA substrate moves faster between active sites than the cleavage rate. Therefore, the RNA binding affinity at the channel entrance, which is in the nM range[41], appears primarily responsible for conferring processivity rather than the binding at active sites. Of note, the influence per se of the number of sites on processivity vs. distributivity remains to be tested experimentally for the archaeal exosome. The outcome of such experiments might vary between distinct RNA substrates according to their respective binding affinity for the exosome. To investigate such features for the plant exosome and to experimentally determine the precise RNA path(s) leading to

RRP41's active site will require the reconstitution of a plant exosome from recombinant proteins, which has failed for technical issues so far. The affinity-purification strategy we employed in this study is not suited to perform biochemical experiments requiring large amounts of proteins. However, its main advantage is that a native, albeit tagged, exosome purified from plants can be studied. By comparing RRP41$^{WT}$-Exo9 with two Exo9s inactivated by the incorporation of the RRP41$^{Pi-}$ or RRP41$^{Pi-Cat-}$ catalytic mutants, we solve a longtime issue about the potential activity of the core exosome in plants.

We also show that the plant Exo9's activity contributes to bona fide functions of the exosome: the elimination of the 5′ ETS excised from the 18S-5.8S-25S rRNA primary transcript and 5.8S rRNA processing. Exo9's phosphorolytic activity has either specific roles (e.g. during 5′ ETS degradation) or acts redundantly with RRP44 and RRP6L2, the two other ribonuclease activities coordinated by the exosome in the plant nucleolus. The occurrence of novel maturation intermediates when exosome function is impaired reveals the existence of alternative ribonucleases also acting on both P-P′ fragments and 5.8S rRNA precursors. Despite this redundancy of ribonucleolytic activities, our 3′RACE-seq strategy identified P-P1 degradation intermediates that are specific for Exo9's activity. Combined with data from previous studies, a precise path for the elimination of the P-P′ fragment generated from the 1.8 kb long 5′ ETS in *Arabidopsis* is emerging. The 481 nt P-P′ fragment is excised by two endoribonucleolytic cuts[13,33]. Its exoribonucleolytic degradation involves the exosome and its cofactors RRP44, RRP6L2, and MTR4[11,13,34,35]. The 3′–5′

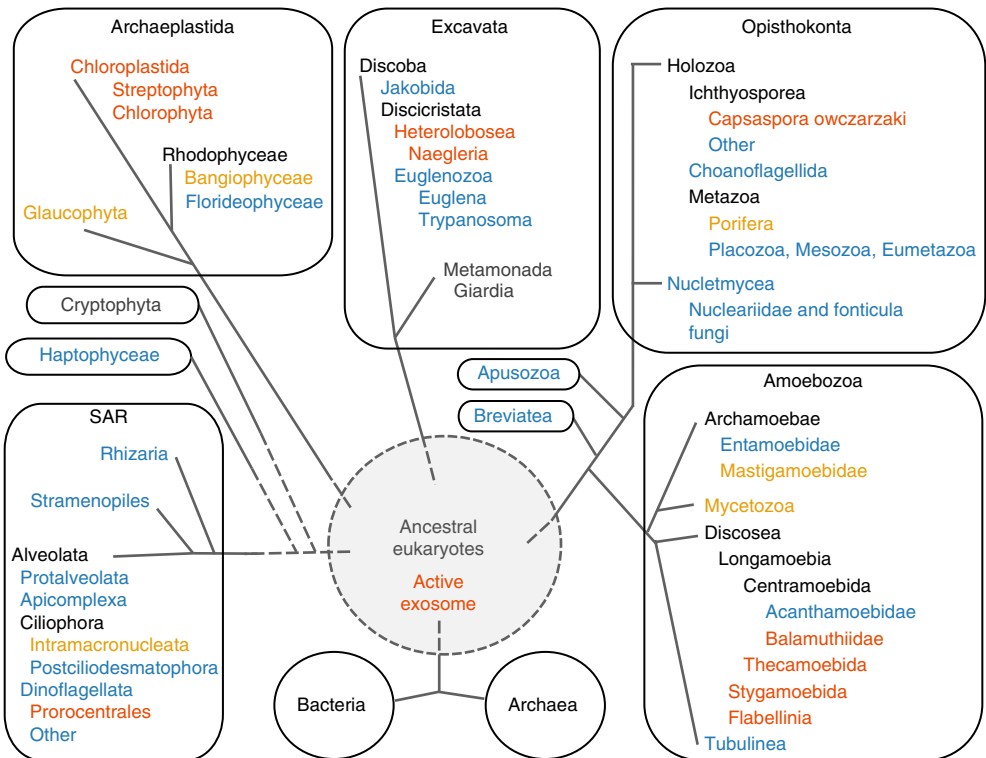

**Fig. 8** Presence of potentially active Exo9 in eukaryotes. Sequences of RRP41 proteins from organisms of all eukaryotic phyla were analyzed for the conservation of amino acids that confer phosphorolytic activity to the *Arabidopsis* exosome. Red letters indicate that all species examined likely possess an active Exo9 complex based on the strict conservation of necessary amino acids in the respective RRP41 orthologues. Orange letters indicate a partial conservation of RRP41 amino acids, i.e. the substitution of critical residues by amino acids with similar chemical properties. Blue letters indicate species whose RRP41 proteins are predicted to be catalytically inactive. Bacteria or Archaea were not included in this analysis as they do not posses the nine-subunit eukaryotic exosome complex per se. However, many of both Archaea and Bacteria possess phosphorolytic active enzymes that share structural similarity with eukaryotic Exo9

degradation of the P-P′ fragment, and many of its subsequent degradation intermediates, is triggered, or at least facilitated, by polyadenylation. As previously shown, the terminal nucleotidyltransferase TRF4/5-LIKE (TRL) is mandatory for the efficient degradation of specific P-P′ intermediates[14]. While those particular intermediates accumulate only in a *trl* mutant background, several P-P1 degradation intermediates of up to 186 nt (+1 corresponds to the P site) are detected even in wild type *Arabidopsis*. Interestingly, a recent global analysis of the RNA–protein interaction sites and RNA secondary structures of *Arabidopsis* nuclear RNA[42] revealed a protein-protected site (PPS) precisely in this area (from nucleotides 85 to 180). Therefore, the four main P-P1 degradation intermediates detected by 3′RACE-seq (P161, P168, P176, and P186) likely reflect the stalling of the exosome at this PPS. Importantly, the P161 and P176 intermediates are absent when Exo9 is inactivated, strongly suggesting that Exo9 produces these intermediates by nibbling P168 and P186, respectively. In line with this, the major P-P1 intermediate produced in absence of MTR4 is P176 when Exo9 is active and P186 when Exo9 is inactive (Fig. 3f). Because the absence of MTR4 severely decreases the occurrence of the smaller P-P1 intermediates, we conclude that the exosome (presumably due to the processive activity of RRP44) degrades P-P′ until its progression is impeded at position 186. Exo9 is able to nibble this intermediate further to generate P176 and this nibbling does not require MTR4. P176 becomes a substrate of RRP6L2 to generate P168 (as deduced from the respective intensities of P176/P168 in Fig. 3c and d). For yet unknown reasons, only Exo9 can further nibble P168 to produce P161. Altogether, our data indicate that three exoribonucleolytic

activities are coordinated by the exosome to repeatedly attack P-P1 degradation intermediates. These combined attacks by both phosphorolytic and hydrolytic activities, facilitated by the polyadenylation of two of the main P-P1 substrates, ultimately lead to overcome a main obstacle impeding the final elimination of the P-P′ rRNA maturation by-product.

Residues compatible with RRP41's activity are remarkably conserved in plants, indicating a certain evolutionary pressure to conserve this trait. However, the sole inactivation of RRP41 is not lethal in *Arabidopsis*. Therefore, Exo9's activity is not essential, albeit its assembly is. Yet, the biological importance of Exo9's activity is revealed when the other catalytic activities linked to exosome function are impaired, and plants are grown under conditions maximizing growth of individual leaves. Clearly, future work is required to fully understand the impact of Exo9's activity on plant phenotypes, especially in the context of adverse conditions. The exhaustive identification of targets of Exo9's phosphorolytic activity will also help to understand the reason for its widespread conservation in the plant lineage. It would be extremely interesting to determine whether Exo9's activity participates in the degradation of all typical classes of exosome substrates that including various non-coding RNAs and mRNAs. However, such a transcriptome-wide identification will be complicated by the fact that at least some specific targets of Exo9 have only a few nucleotide extensions. Another aspect to consider in order to establish the full biological roles of Exo9's activity is that Exo9 might also tail RNAs. The phosphorolytic reaction is almost thermodynamically neutral, and therefore phosphorylases can either degrade or tail RNA substrates. Such tailing has been

demonstrated in vivo for plastidial and bacterial PNPases and for the archaeal exosome[6,43–47]. Therefore, the likelihood that the plant exosome is able to tail RNAs in vivo definitely exists. This would constitute an interesting peculiarity in the metabolism of plant nucleus-encoded RNAs amongst eukaryotes.

In conclusion, we show that *Arabidopsis* Exo9 has a unique distributive phosphorolytic activity conferred by RRP41, which is potentially conserved in most representatives of the green lineage. In *Arabidopsis*, Exo9s activity contributes, alongside with RRP6L2's and RRP44's hydrolytic activities, to the removal of rRNA maturation by-products and 5.8S rRNA maturation. We propose that the co-existence of RRP6L2's and RRP44's hydrolytic activities with RRP41's phosphorolytic activity in *Arabidopsis* confers robustness to the action of the RNA exosome. In addition, our results challenge the idea that RNA degradation in the cytosol and nucleus of eukaryotes is only hydrolytic. Our data also imply that the plant RNA exosome has a unique combination of hydrolytic and phosphorolytic ribonucleolytic activities, making this complex a remarkably versatile RNA degradation machine.

## Methods

**Plant materials**. All *A. thaliana* plants were of Columbia ecotype (Col-0). The T-DNA insertion lines *rrp6L2*, and *mtr4*, and amiRRP44 lines downregulated for RRP44 expression have been described previously[13,34,35]. Plants expressing MTR4-GFP were described in ref. [34]. Plants expressing PAB2-RFP were a kind gift of Cécile Bousquet-Antonelli and Jean-Marc Deragon (Perpignan, France). For RRP4-GFP plants, the *RRP4* cDNA was obtained by RT-PCR and cloned into a pUCAP plasmid already containing the cauliflower mosaic virus (CaMV) 35S promoter, an E-tagged version of an *eGFP* gene and the CaMV 35S terminator. The expression cassette was then transferred into pBinH, a pBINPLUS derivative with a hygromycin resistance marker[48]. The resulting binary vector allowing the expression of RRP4-GFP fusion proteins under the control of the CaMV 35S promoter was used for agrobacterium-mediated transformation of Col-0 plants. *rrp41* RRP41^WT, *rrp41* RRP41^Pi-, and *rrp41* RRP41^Pi-Cat- plants were generated as follows: the genomic sequence of the *RRP41* gene including 2 kb upstream of the ATG codon comprising the putative endogenous promoter was amplified from genomic DNA and cloned into pDONR207 using the Gateway technology. Point mutations in RRP41^Pi- and RRP41^Pi-Cat constructs were introduced by PCR and generated diagnostic *Nde*I and *Bsr*I restriction sites, respectively. In RRP41^Pi-, residues in the phosphate coordination pocket were changed (R131Y, S132A) in order to mimic the respective residues of the catalytically inactive human RRP41 protein. For RRP41^Pi-Cat- both the phosphate coordination site and a conserved aspartate required for magnesium coordination in the catalytic center were mutated (R131Y, S132A, D174A). RRP41^WT, RRP41^Pi-, and RRP41^Pi-Cat- sequences were cloned into pGWB604 and pGWB616[49] and the resulting binary vectors were used for agrobacterium-mediated transformation of heterozygous RRP41/rrp41 (Salk_112819) plants using the floral dip method[50]. Homozygous *rrp41* mutants expressing either myc-tagged or GFP-tagged RRP41^WT, RRP41^Pi-, and RRP41^Pi-Cat- proteins were identified in the selfed progeny of primary transformants using PCR and restriction analysis. For double mutants lines, *rrp41* RRP41^WT, *rrp41* RRP41^Pi-, and *rrp41* RRP41^Pi-Cat plants were crossed with *rrp6L2* or *mtr4* mutants, and double homozygous plants were isolated from the selfed F2 generation. Downregulation of RRP44 expression was achieved by transforming Col-0, *rrp41* RRP41^WT, *rrp41* RRP41^Pi-, *rrp41* RRP41^Pi-Cat, *rrp6L2*, *rrp6L2 rrp41* RRP41^WT, *rrp6L2 rrp41* RRP41^Pi-, and *rrp6L2 rrp41* RRP41^Pi-Cat plants with pFASTG0-NK14, which drives the expression of an artificial miRNA directed against RRP44[35] under the control of the mesophyll-specific *CAB3* promoter from a binary plasmid allowing the selection of primary transformants by green seed technology[51]. Primary transformants were tested for the accumulation of the excised 481 nt P-P′ fragment of the 5′ ETS as an biological indicator for efficient RRP44 downregulation[35], and plants with comparable levels of P-P′ accumulation were selected for proliferation.

**Plant growth conditions**. Plants used for gel-filtration, immunopurifications, and activity assays, as well as the plants whose phenotypes are shown in Fig. 6a and Supplementary Fig. 6b–d were sown on soil, grown for 2 weeks in 12 h light (75–90 µmol s$^{-1}$ m$^{-2}$)/12 h darkness with Cool Daylight HO 49W/865 T5 tubes, outbedded to individual pots, and grown for further 4 weeks (rosette leaves) or 6 weeks (flowers) in 16 h light (ca. 140 µmol s$^{-1}$ m$^{-2}$)/8 h darkness with Cold white Master 20W/840 T8 tubes. For 3′RACE-seq and Northern blot analysis, we chose conditions that minimize eventual secondary effects conferred by the developmental phenotypes of double and triple mutants. Therefore, plants were directly sown in individual pots and cultured for 6 weeks in 12 h light (ca. 110 µmol s$^{-1}$ m$^{-2}$)/12 h darkness under illumination with Cool Daylight HO 49W/865 T5 tubes. Plants grown under these conditions are shown in Supplementary Fig. 6a. Plants used for

microscopy and cotyledon analysis were grown in vitro on Murashige and Skoog solid medium supplemented with 0.5% sucrose in 16 h light (from 100 to 140 µmol s$^{-1}$ m$^{-2}$)/8 h darkness illuminated with Biolux L 58W/865 T8 or Cold white Master 20W/840 T8 tubes. We also examined cotyledons of plant grown in soil under similar light/dark conditions, but did not observe any differences to plants grown in vitro.

**Structure modeling**. The structure of the wild type archaeal *S. solfataricus* exosome (PDB ID: 3l7Z)[26] was aligned using the PyMOL software to the structure of the *S. solfataricus* exosome with mutated RRP41 (D182A) and bound to inorganic phosphate (PDB ID: 4BA1)[30]. This alignment helped to identify the position of the phosphate ion linked to Rrp41 in the WT context. The structure obtained was aligned to the structure of *E. coli* PNPase (PDB ID: 3GME)[28]. This second alignment helped to position a magnesium ion in the catalytic site of Rrp41. AtRRP41 (AT3G61620) and AtRRP45A (AT3G12990) structures were modeled with high confidence against human, yeast, and archaeal exosome structures using Phyre2[52]. The modeled structures were aligned to the structure of *S. solfataricus* Rrp41–Rrp42 bound to inorganic phosphate and metal ion to visualize the residues potentially implicated in RNA binding and in the coordination of Pi and Mg$^{2+}$. Uniprot accessions are: At_41 (Q9SP08); At_45A (Q9LDM2); At_45B (Q9M209); Ss_41 (Q9UXC2); Ss_42 (Q9UXC0); Pa_41 (Q9V119); Pa_42 (Q9V118); Af_41 (O29757); Af_42 (O29756); Mt_41 (O26779) and Mt_42 (O26778).

**Analysis of RRP41 protein diversity**. Analysis of RRP41 protein diversity was performed essentially as described in ref. [53]. RRP41 sequences were retrieved using BLAST[54] or HMMER[55] using either the *Arabidopsis* RRP41 sequence or a Rrp41 sequence of the concerned phylogenetic group as query. The orthologues were then used as template in a BLAST against *Arabidopsis* proteome (TAIR) to confirm their homology to AtRRP41 vs. the closely related subunits AtRRP46 or AtMTR3. Full length sequences of RRP41 from the sub-groups specified in Fig. 7 were aligned using Muscle[56]. Sequence logos were calculated with WebLogo 3[57]. Sequences and accession numbers are provided in Supplementary Datas 1 and 2.

**Confocal microscopy**. The intracellular localization of GFP fusion proteins was determined in three independent transformed lines of each *rrp41* RRP41^WT, *rrp41* RRP41^Pi-, and *rrp41* RRP41^Pi-Cat-. Plants expressing GFP-RRP4, GFP-MTR4, and RFP-PAB2 were used as controls. Plants were grown on MS agar plates supplemented with 0.5% sucrose. Roots from 10-day-old seedlings were excised, placed with water under a coverslip and examined with a ZEISS LSM 780 confocal microscope.

**Determination of vascular leaf patterning in cotyledons**. Cotyledons of 7–10-day-old seedlings were placed in water containing 0.0025% Tween 20, covered with a coverslip, vacuum-infiltrated for 3 min to remove air from the leaf tissue and observed with a stereo microscope. Vein patterns were classified as "aberrant "when the two early appearing veins failed to form the closed structure that is usually observed in wild type cotyledons.

**Gel filtration**. Two hundred milligrams of flowers from *rrp41* myc-RRP41^WT or *rrp41* myc-RRP41^Pi-Cat- transgenic lines were ground in 1 ml of 20 mM Tris–HCl pH 7.6, 150 mM NaCl, 0.5% Tween 20, EDTA-free cOmplete™ protease inhibitors (Sigma Aldrich) at 4 °C. Crude extracts were clarified by centrifugation at 16,000×*g* and 150,000×*g* for 5 and 15 min, respectively. Two hundred and fifty microliters of the ultracentrifugation supernatant were analyzed by gel filtration using a Superose 6 20 10/300 column (GE Healthcare Life Sciences) equilibrated in 20 mM Tris–HCl pH 7.6, 150 mM NaCl, 0.1% Tween 20 and run at 0.25 ml/min. Elution fractions containing myc-tagged RRP41 subunits were identified by western blot analysis using a anti-myc monoclonal antibody (Roche). The Superose 6 column was calibrated using Thyroglobulin (669 kDa), Aldolase (158 kDa) and RNaseA (14 kDa) markers (GE Healthcare Life Sciences).

**Protein purification**. For immunopurification of Exo9, 200 mg of flowers from Col-0 or myc-tagged *rrp41* RRP41^WT, *rrp41* RRP41^Pi-, or *rrp41* RRP41^Pi-Cat- plants were ground in liquid nitrogen in lysis buffer (50 mM Tris–HCl pH 8.0, 150 mM NaCl, 1% Triton X-100, EDTA-free cOmplete™ protease inhibitors). All following procedures were performed at 4 °C. Crude extracts were clarified by centrifugation at 16,000×*g* and 150,000×*g* for 5 and 15 min, respectively. After 30 min of pre-incubation with anti-myc antibodies coupled to magnetic MicroBeads (Miltenyi Biotec), samples were bound to magnetized MACS separation columns equilibrated with lysis buffer. Columns were rinsed four times with lysis buffer, twice with 20 mM MOPS pH 7.5, 250 mM NaCl, 0.1% Triton X-100 and twice with 20 mM MOPS pH 7.5, 100 mM NaCl, 0.1% Triton X-100. Samples were collected in 20 mM MOPS pH 7.5, 100 mM NaCl, 0.1% Triton X-100 by removing columns from the magnetic field. Glycerol was added to a final concentration of 10% and samples were frozen in liquid nitrogen and stored at −80 °C.

A similar protocol was used to immunopurify Exo9 containing RRP4-GFP except that flowers from RRP4-GFP lines and anti-GFP antibodies coupled to magnetic MicroBeads (Miltenyi Biotec) were used.

**Mass spectrometry**. Mass-spectrometric analysis was carried out as described[12]. Samples were eluted from magnetic beads in Laemmli buffer, precipitated with 100 mM ammonium acetate in methanol, and resuspended in 50 mM ammonium bicarbonate. After reduction and alkylation steps with 5 mM dithiothreitol and 10 mM iodoacetamide, respectively, proteins were digested overnight with trypsin 1/25 (w/w). Vacuum-dried peptides were dissolved in 15 µl 0.1% formic acid (solvent A). One-third of each sample was injected on a NanoLC-2DPlus system (nanoFlex ChiP module; Eksigent, ABSciex, Concord, Ontario, Canada) coupled to a TripleTOF 5600 mass spectrometer (ABSciex) operating in positive mode. Peptides were loaded on C18 columns (ChIP C-18 pre-column 300 µm ID × 5 mm ChromXP and ChIP C-18 analytical column 75 µm ID × 15cm ChromXP; Eksigent) and were eluted using a 5–40% gradient of solvent B (0.1% formic acid in acetonitrile) for 60 min at a 300 nl/min flow rate. The TripleTOF 5600 was operated in high-sensitivity data-dependent acquisition mode with Analyst software (v1.6, ABSciex) on a 350–1250 $m/z$ range. Up to 20 of the most intense multiply charged ions (2+ to 5+) were selected for CID fragmentation, with a cycle time of 3.3 s (TOP 20 discovery mode).

For protein identification, raw data were converted to Mascot Generic File format (mgf) and searched against a TAIR 10 database supplemented with a decoy database build from reverse sequences. Data were analyzed using Mascot algorithm version 2.2 (Matrix Science, UK) through ProteinScape 3.1 software (Bruker). Search parameters allowed N-acetylation (protein N-terminal), carbamidomethylation (C) and oxidation (M) as variable peptide modifications. Mass tolerances in MS and MS/MS were set to 20 ppm and 0.5 Da, respectively. 2 trypsin mis-cleavages were allowed. Peptide identifications obtained from Mascot were validated with a FDR < 1%. Identified proteins were assessed by the total number of fragmented spectra per protein (spectral count).

**In vitro activity assays**. RNA substrates were a 5′[$^{32}$P]-labeled oligo(U)$_{21}$ in Fig. 2 and Supplementary Fig. 3 or a 21 nt 5′[$^{32}$P]-UCGCUUGGUGCAGGUCGGGAA-3′ in Supplementary Fig. 2. For TLC analysis, oligo(U)$_{21}$ RNA was 3′ labeled with [$^{32}$P]αUTP using Cid1 poly(U) polymerase (New England Biolabs). Exo9 concentration was 1.4 nM in Fig. 2c–e and Supplementary Fig. 2, or 0.4 nM in Fig. 2b, respectively. RNA concentrations were 6 nM in Fig. 2b, 25 nM Fig. 2c–e and Supplementary Fig. 2. Activity assays were carried out at 20 °C in 20 mM MOPS pH 7.5, 50 mM NaCl, 1.5 mM MgCl$_2$, 1.5 mM DTT, 1 U/µl RNase inhibitor, 0.05% Triton X-100. +Pi assays contained 3.5 mM potassium phosphate. Synthesis assays contained 1 mM of UDP. Aliquots were taken at indicated time points and the enzymatic reaction was stopped by adding 1 volume of formamide supplemented with 10 mM EDTA, 0.1% bromophenol blue, and 0.1% xylene cyanol FF. Samples were separated on 16% polyacrylamide, 7 M urea and TBE (220 mM Tris, 180 mM borate, 5 mM EDTA pH 8.3) and visualized by autoradiography. For TLC analysis, reactions were stopped with 10 mM EDTA before samples were loaded on PEI-Cellulose F plates (Merck Millipore), resolved in 0.5 M LiCl 1 M formic acid, and visualized by autoradiography. Identical results were obtained with exosomes purified from four biological replicates, each replicate consisted of pooled flowers from 10 plants.

**Northern blot analysis**. RNA was extracted from rosette leaves of indicated genotypes using TRI Reagent® (MRC). Five micrograms of total RNA were separated on 6% polyacrylamide, 7 M Urea and TBE, stained with ethidium bromide and transferred to Hybond™-N+ (GE Healthcare Life Sciences™). For northern blots, membranes were equilibrated for 30 min in 0.5 M sodium phosphate pH 7.2, 7% SDS, hybridized at 42 °C overnight with 5′ [$^{32}$P] - labeled oligonucleotides O4-6 (Supplementary Table 3) and analyzed by autoradiography.

**3′RACE-seq library preparation**. 3′ ends of rRNA precursors and rRNA maturation by-products were determined by a modified 3′RACE procedure adapted to Illumina high throughput sequencing. RNA was extracted from rosette leaves using TRI Reagent (MRC). Fifty picomoles of a 5′-riboadenylated DNA oligonucleotide comprising a 5 nt delimiter, 15 nt of randomized sequence, and 22 nt reverse complement to the reverse Illumina TruSeq RNA PCR Index primers (oligonucleotide 3′-Adap, Supplementary Table 3) was ligated to 20 µg of total RNA using T4 RNA ligase 1. Ligated RNA was separated by electrophoresis in 6% polyacrylamide, 7 M urea, TBE. RNA molecules of 100–400 nt were excised and eluted overnight at 4 °C in 200 µl 0.5 M ammonium acetate, 10 mM magnesium acetate, 1 mM EDTA, and 0.1% SDS. After extraction with phenol–chloroform, RNA was precipitated with ethanol, and solubilized in water. Five hundred nanograms of size-selected RNA samples were used in 25 µl cDNA synthesis reactions each containing 1 U of Superscript IV (Thermo Fisher Scientific), 2 µM of a primer complementary to the last 17 nt of the ligated adapter sequence (oligonucleotide 3′-RT, Supplementary Table 3), and 10 mM dNTPs in the buffer supplied by the manufacturer.

Libraries were amplified by PCR using Dreamtaq polymerase (Thermo Fisher Scientific). Forward primers comprised the Illumina P5 sequence and 21 nt of the 5′ ETS downstream of the P processing site (oligonucleotide O1, Fig. 3a and Supplementary Table 3), or 21 nt of either mature or 3′ extended 5.8S rRNA (oligonucleotides O2 and O3, Fig. 5a and Supplementary Table 3). Reverse primers were TruSeq RNA PCR index primers (RPI, Illumina). Libraries were purified with

1 volume of Agencourt AMPure XP beads (Beckman coulter) and paired-end sequenced with MiSeq (v3 chemistry) with 41 × 101 bp cycle setting. To compensate for the poor diversity of the rRNA amplicon libraries, 16% of phiX control library (Illumina) was included. For each genotype, two independent biological replicates were analyzed.

**3′RACE-seq data processing**. After initial data processing by the MiSeq Control Software v 2.5. (Illumina), base calls were retrieved and further analyzed by a suite of home made scripts (Supplementary Data 3) using python (v2.7) and biopython (v1.63) and regex (v2.4) libraries. Sequences with identical nucleotides in read 1 (insert) and the 1st to 15th cycle in read 2 (randomized bases in 3′ adapter) were deduplicated. Next, the sequences TTCTGGCCGAGGGCACGTCTG for 5.8S mature rRNA, TCTGCCTGGGTGTCACAA for 5.8S rRNA precursors and ATCTCGCGCTTGTACGGC for the P-P1 fragments of the 5′ ETS were searched in reads 1. One mismatch was allowed. Reads 1 that did not match were discarded. The remaining reads 1 were extracted and annotated alongside with their corresponding read 2. Next, reads 2 that did not contain the delimiter sequence were discarded. For the remaining reads 2, the randomized sequence and the delimiter present in the 3′ adapter were removed. As the target sequences were often short, reads 2 often run into the 5′ PCR primer, the sequences of which were also removed. Reads 2 shorter than 20 nt after these trimming steps were excluded from further analysis. In order to determine the 3′ extremities of the target RNAs, reads 2 were mapped to the reference sequence of the A. thaliana rRNA repeat unit (Genbank X52322.1), with 3′ end position 1 corresponding to the first nucleotide of the 5.8S or 5′ ETS sequence that matches the 5′ PCR primer. rRNA species that contain non-templated tails produce reads 2 that do not match the reference sequence. To identify such reads and map their 3′ end position, the sequences of the unmatched reads 2 were successively trimmed from their 3′ end. After each trimming step of 1 nt, sequences were mapped to the reference genome, and the unmatched sequences were further trimmed until a maximum of 30 nt has been removed. At each mapping step, non-templated nucleotides at the 3′ were extracted and analyzed for their size and composition. If the second nucleotide matched to the reference sequence, the corresponding read was not considered as 3′ modified and removed from the analysis. Furthermore, 3′ modifications longer than 6, 10 and 15 nt were considered only if they contained at least one stretch of AAA or TTT, two stretches of AAA or TTT, or three stretches of AAA or TTT, respectively. Final processing was performed with R software (v3.3.1). Reads mapping upstream position of the 5′ PCR primer were discarded and position of 3′ end extremities were corrected for 5.8S rRNA, so that position 1 corresponds to the first nucleotide after the end of the 5.8S mature form.

**3′RACE-seq plots**. Plotting and quantitative data analysis was performed with R software (v3.3.1) and ggplot2 R (v2.2.1).

**Statistical analysis**. If not stated otherwise, a biological replicate is defined as plants of the same genotype grown under similar conditions but at different times. For each genotype, RNA or protein was extracted from a pool of 5–10 individual plants.

The 3′RACE-seq analysis was performed with two biological replicates that were used for the construction of independent amplicon libraries. Each of the two biological replicates was analyzed separately except for the analysis of the tail length, where data from the two replicates were compiled (Fig. 4c). In this figure, the median is indicated by the black bars, the first and third quartiles by the box edges, the inter-quartile range by thin bars and the outliers by dots and the data range (except outliers) is shown by whiskers.

In vitro activity assays and northern blots were performed with at least three biological replicates.

Gel filtration experiments were performed once with proteins extracted from myc-tagged and once with protein extracted from GFP-tagged rrp41 RRP41$^{WT}$ and rrp41 RRP41$^{Pi-Cat-}$ plants. Mass-spectrometric analysis of purified exosome complexes was performed with one replicate consisting of Col-0, rrp41 RRP41$^{WT}$, rrp41 RRP41$^{Pi-}$, and rrp41 RRP41$^{Pi-Cat-}$. Each of the myc-tagged baits co-purified with similar amounts of the other Exo9 subunits and results were in full agreement with previously published results[12].

Vascular patterns in cotyledons were analyzed in two biological replicates comprising 25–100 individual plants each. A Pearson's Chi-squared test with Yates' continuity correction was used to assess whether the frequency of aberrant vein patterns between genotypes were statistically different. For this statistical analysis, data from biological replicates 1 and 2 were pooled.

**Gel and blot images**. Full uncropped versions of all gel/blot images are shown in Supplementary Fig. 8. All raw images are available at figshare with the DOI 10.6084/m9.figshare.5509504.

**Data availability**. The 3′ RACE-seq and mass spectrometry data that support the findings of this study have been deposited to the NCBI Gene Expression Omnibus (GEO) database with the accession code GSE96948 and to the ProteomeXchange Consortium via the PRIDE[58] partner repository with the dataset identifier PXD007984, respectively.

Raw data are available at figshare with the doi: 10.6084/m9.figshare.5509504.

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

## Acknowledgements

We thank Nicolas Baumberger for assistance with gel filtration and Johana Chicher, Lauriane Kuhn, and Philippe Hammann for nanoLC–MS/MS analyses (Strasbourg-Esplanade Proteomic Facility, IBMC, CNRS, FRC1589). This work was funded by the Centre National de la Recherche Scientifique (CNRS, France) and the French National Research Agency as part of the program d'Investissements d'Avenir in the frame of the LABEX (ANR-2010-LABX-36 to D.G.). The funders had no role in study design, data collection and analysis, decision to publish, or preparation of the manuscript.

## Author contributions

D.G. and H.L. conceived and designed the study; N.S., H.Z., and H.L. performed experiments; H.Z. performed bioinformatics analysis; A.G. analyzed RRP41 protein diversity; D.G. and H.L. wrote the paper; N.S., H.Z., and A.G. edited the manuscript; N.S., H.Z., A.G., and H.L. prepared illustrations; D.G. acquired funding.

## Additional information

**Competing interests:** The authors declare no competing financial interests.

