## [Peer Review File · Nature Communications]

Reviewers' comments:

Reviewer #1 (Remarks to the Author):

In this paper the authors address a long-standing question regarding the RNA exosome. The bacterial and archaeal equivalents have phosphorylase activity that is critical for their function. The yeast and animal exosomes do not have this catalytic activity. Finally, based on some not convincing biochemistry it has long been speculated that the plant exosome has catalytic activity (ref 19). In this work the authors definitively show that this is indeed the case and identify some of the molecular functions that this activity is required for. This is a beautiful set of investigations that without a doubt deserves publication in Nature Communications as is. I expect this to be a highly cited paper.

The authors may want to consider the following very minor points:

1. If the plant RNA exosome retains catalytic activity, this suggests that a common ancestor of animals and fungi lost the activity after divergence from plants. A corollary of that is that earlier diverging eukaryotes, which include important human pathogens like trypanosome, Giardia and Plasmodium, may also have retained exosome activity. The authors might want to point this out.
2. The paper is easily understood by an exosome aficionado, but the authors might want to tweak the text for a broader audience.
3. Line 77 "Amino acids critical for catalysis belong to Rrp41 but are located near the interface of Rrp41-Rrp42 dimers." I think the authors are discussing the archaeal situation here. Can they clarify?
4. Line 235 "when either RRP6L2 or RRP44 are impaired" should be "when either RRP6L2 or both RRP6L2 and RRP44 are impaired"
5. Line 335 discusses the absence of an obvious phenotype of the rrp41-Pi and rrp41-Pi-Cat mutants. The authors should also comment of the phenotype of the double and triple mutants where rrp41 is combined with rrp6-l2, rrp44, or mtr4.
6. Line 79, rationale should be rational.
7. Lines 411 and 413, "Blast" should be "BLAST"

Reviewer #2 (Remarks to the Author):

The manuscript by Sikorska et al., investigates role of the 3' to 5' phosphorylase activity associated with the core subunit of the plant RNA exosome Rrp41, which is catalytically inactive in other eukaryotes. The study builds upon previous work from Belostotsky's lab demonstrating phosphorylase activity for recombinant Rrp41 and now demonstrates distributive phosphorylase activity in the context of the nine-subunit core (exo9) of the complex and proposes that phosphorylase activity of the core complex complements hydrolytic activities of the exo9+RRP6 or DIS3 (RRP44), which is supported by additive accumulation of byproducts of rRNA processing when both activities (hydrolytic and phosphorylase) are abolished. Interestingly, similar to archaeal homologue, Arabidopsis RRP41 also possesses terminal transferase activity to RNA, however in contrast to archaeal enzyme, which is a processive exonuclease, it shows distributive mode of action. It is a nice simple story, overall data seem to be of high quality but it appears to me that the insights into how phosphorylase activity contributes to the function of the exosome as well as advance over published work are somewhat

limited and therefore may not be suitable for the publication in Nature Communications in its current form.

Specific points:

1. I am not quite convinced based on the data presented of the functional importance of the phosphorolytic activity. The single RRP41 mutant shows no effect on processing of rRNA (fig 5b). There is some additive effect seen in combination with RRP6, but the phosphorolytic activity seems somewhat redundant with hydrolytic enzymes. Are there RNA targets that are specifically dependent on phosphorolytic activity? Is there any effect on the overall fitness of the plant upon loss of RRP41 activity?
2. Relationship between hydrolytic and phosphorolytic activities should be investigated in more details in vitro and in vivo in order to understand their contributions.
3. I am not convinced that rRNA processing is affected rather than degradation of the precursor molecule in Fig 5. Is there simultaneous depletion of mature rRNA species (should be included)?
4. For convenience of the readers, Lanes should be labelled in all the figures and referenced in the text. Diagrams describing rRNA processing and the intermediates detected by Northern should be included.

Reviewer #3 (Remarks to the Author):

This manuscript reports that the Arabidopsis exosome nine-subunit core (Exo9) has a distributive phosphorolytic activity and participates in the degradation of RNA byproducts and processing rRNA precursors during rRNA maturation. The authors show that Rrp41 in Exo9 is responsible for Exo9's phosphorolytic activity and that Exo9 coordinates with Rrp6L2 and Rrp44 for RNA processing during rRNA maturation. This manuscript thus reports a novel finding that is supported by solid biochemical data. However, some concerns listed below are required to be addressed.

1. Rrp41 forms a dimer with Rrp42 in yeast/human exosome, but not Rrp45. It is not clear why the structure model shown in Figure 1c is made for Arabidopsis Rrp41-Rrp45, but not for Rrp41-Rrp42? Figure 1 is also not clear. For example, the Mg²⁺ ion is not shown in the model. The phosphate binding residues are not marked in the sequence alignment (Figure 1a and 1b).
2. It has been shown that the rice Rrp46 forms a homodimer bearing phosphorolytic activity (RNA 16: 1748-1759, 2010). The authors need to compare the sequence and activity of Arabidopsis Rrp41 with that of rice Rrp46. Is it possible that the rice Rrp46 was actually the homolog of Arabidopsis Rrp41? Does Arabidopsis Rrp41 form a homodimer for RNA degradation (that probably can explain its distributive enzymatic activity)? How do we know that the phosphorolytic activity observed in Figure 2 did not result from Rrp41 alone as the protein samples used in this study were purified only by a myc-antibody against myc-tagged Rrp41?
3. The major function of exosome is mRNA degradation. The authors show that the Arabidopsis Exo9 participates in rRNA maturation, however, this manuscript did not answer or discuss the most significant question, if the Arabidopsis Exo9 is involved in mRNA degradation.

Answers to Reviewers' comments:

Changes in the main text are marked in red. Figures have been modified according to Reviewers' comments or to fit Nature Communications guidelines.

Reviewer #1:

In this paper the authors address a long-standing question regarding the RNA exosome. The bacterial and archaeal equivalents have phosphorylase activity that is critical for their function. The yeast and animal exosomes do not have this catalytic activity. Finally, based on some not convincing biochemistry it has long been speculated that the plant exosome has catalytic activity (ref 19). In this work the authors definitively show that this is indeed the case and identify some of the molecular functions that this activity is required for. This is a beautiful set of investigations that without a doubt deserves publication in Nature Communications as is. I expect this to be a highly cited paper.

We thank Reviewer 1 for his/her positive and constructive review. We have addressed two very interesting comments made by Reviewer 1: we investigated the potential existence of a catalytically active exosome in earlier diverging eukaryotes, and we detail the phenotypes of the double mutants. Our revised version includes these novel data and we think that they indeed improve our study.

The authors may want to consider the following very minor points:

1. If the plant RNA exosome retains catalytic activity, this suggests that a common ancestor of animals and fungi lost the activity after divergence from plants. A corollary of that is that earlier diverging eukaryotes, which include important human pathogens like trypanosome, Giardia and Plasmodium, may also have retained exosome activity. The authors might want to point this out.

As suggested by Reviewer 1, we extended our analysis of RRP41 protein diversity to include representative species for the major groups of eukaryotes. Interestingly, we could indeed identify organisms outside the Archaeplastida whose genome encodes a RRP41 subunit which contains all residues required for phosphorylase activity. Such organisms include several Amoebozoa, *Capsaspora owczarzaki* (a close unicellular relative of metazoans) as well as the human pathogen *Naegleria fowleri*. Although experimental validation is clearly beyond the scope of this manuscript, we think that the possibility that RRP41 might be active in non-plant eukaryotes constitutes an interesting point for the discussion of our data on Arabidopsis. Therefore, this novel observation is shown in the new Fig.8 and discussed in the revised version.

2. The paper is easily understood by an exosome aficionado, but the authors might want to tweak the text for a broader audience.

We think that the background information provided in the abstract and introduction is accessible to a wide readership. We agree that some parts of the results and discussion are more for exosome specialists. However we did not identify obvious changes to be made, except that one sentence describing the 35S rRNA precursors in Arabidopsis was added.

3. Line 77 "Amino acids critical for catalysis belong to Rrp41 but are located near the interface of Rrp41-Rrp42 dimers." I think the authors are discussing the archaeal situation here. Can they clarify? We are indeed discussing the archaeal situation. This has been clarified in the text.

4. Line 235 "when either RRP6L2 or RRP44 are impaired" should be "when either RRP6L2 or both RRP6L2 and RRP44 are impaired" □
Modified accordingly.

5. Line 335 discusses the absence of an obvious phenotype of the rrp41-Pi and rrp41-Pi-Cat mutants. The authors should also comment of the phenotype of the double and triple mutants where rrp41 is combined with rrp6-l2, rrp44, or mtr4.

Indeed, when grown under conditions promoting fast growth of leaves and a rapid transition into flowering stage (16h light / 8h darkness), both double and triple RRP41^{Pi- Cat-} mutants show a pronounced growth retardation as compared to their RRP41^{WT} counterparts. All plants used for RNA analysis were grown under conditions promoting slow growth of individual leaves, such as low light intensity and short photoperiods (12h light / 12h darkness). The main reason for this choice was that we knew from previous experience that these conditions minimize the developmental defects of both *mtr4* and *rrp6L2 RR44KD* plants, which limits as much as possible indirect effects of their retarded growth on our molecular analysis. Under these conditions, we did not observe a marked phenotypic effect of mutating RRP41 either alone or in combination with *rrp6L2*, *rrp6L2 44KD* or *mtr4*. Since both

Reviewer 1 and Reviewer 2 commented on this question, we have now analyzed the plants' phenotypes in other growth conditions. We found that when plants are grown with high light intensity and a long photoperiod (16h light / 8h darkness), inactivating Exo9's activity in *rrp6L2*, *rrp6L2 RRP44KD*, or *mtr4* genetic backgrounds has a severe impact on growth.

To support a link between the molecular and the macroscopic phenotypes of the mutant plants, we examined cotyledon vein patterns in all single and double mutants (except the three *rrp6L2 RRP44KD* plants because the promoter which controls the amiRNA to downregulate RRP44 is expressed in mature true leaves). Cotyledon vein pattern formation depends on auxin transport processes in the developing embryo, and is, for reasons not understood yet, typically observed in ribosomal protein and ribosome biogenesis mutants (reviewed in Byrne 2009 and Weis et al. 2015 Trends in Plant Science), including *mtr4* (Lange et al. 2011 Plant J). Interestingly, inactivating Exo9 in *rrp6L2* significantly disturbs vein formation in cotyledons, which mirrors the accumulation of 5.8S rRNA precursors that we observe in *rrp6L2 rrp41* RRP41^{Pi-Cat⁻} plants (see also answers to Reviewer 2 below). Taken together, these new data strongly support our conclusion that Exo9's activity contributes to ribosome biogenesis and further reveal the biological impact of RRP41's activity on plant development.

These new results are detailed in our answers to points 1 & 3 made by Reviewer 2 and the corresponding data have been integrated in the revised version (Fig. 6, Supplementary Fig. 6 and 7). We now provide detailed culture conditions for each type of experiment in Material and Methods. We also noticed that we forgot to detail the growth conditions for the plants used for RNA analysis in the first version of the manuscript. We apologize for this omission.

6. Line 79, rationale should be rational.

Corrected.

7. Lines 411 and 413, "Blast" should be "BLAST"

Corrected.

Reviewer #2:

The manuscript by Sikorska et al., investigates role of the 3' to 5' phosphorolytic activity associated with the core subunit of the plant RNA exosome Rrp41, which is catalytically inactive in other eukaryotes. The study builds upon previous work from Belostotsky's lab demonstrating phosphorolytic activity for recombinant Rrp41 and now demonstrates distributive phosphorolytic activity in the context of the nine-subunit core (exo9) of the complex and proposes that phosphorolytic activity of the core complex complements hydrolytic activities of the exo9+RRP6 or DIS3 (RRP44), which is supported by additive accumulation of byproducts of rRNA processing when both activities (hydrolytic and phosphorolytic) are abolished. Interestingly, similar to archaeal homologue, Arabidopsis RRP41 also possesses terminal transferase activity to RNA, however in contrast to archaeal enzyme, which is a processive exonuclease, it shows distributive mode of action. It is a nice simple story, overall data seem to be of high quality but it appears to me that the insights into how phosphorolytic activity contributes to the function of the exosome as well as advance over published work are somewhat limited and therefore may not be suitable for the publication in Nature Communications in its current form.

We are pleased that Reviewer 2 judges that our "overall data seem to be of high quality" and found that our manuscript reports a "nice simple story". Yet, we would like to clarify our position regarding two points he/she makes. The first remark we do not agree with is that the Chekanova/Belostotsky lab demonstrated the phosphorolytic activity for recombinant RRP41. In agreement with Reviewer 1, we think that this conclusion is "based on some not convincing biochemistry". Reporting a poly(A)-dependent activity based on a recombinant protein purified from *E. coli* and with no activity mutant cannot be considered as a "demonstration" nowadays. Using a similar strategy, the same authors also reported an exoribonucleolytic activity for RRP4 (Chekanova et al 2002 NAR), while the sequence comparisons, structural data and biochemical results that are available today show that both eukaryotic and archaeal RRP4 proteins are RNA binding proteins completely devoid of catalytic activity. We also note that we did not observe the poly(A) dependence for RNA degradation by Exo9 that was reported for recombinant RRP41. Of course we cannot formally rule out that this could be due to the fact that RRP41 is embedded in Exo9. However, this discrepancy reveals some of the limits for the original conclusion of Chekanova et al. Even if RRP41 would be active as a monomer (in contradiction to the situation observed for archaeal Rrp41 which is active only as a heterodimer with

Rrp42), there is little biological meaning in this observation since we did not observe monomeric RRP41 *in vivo*. Therefore, our study is the first to really demonstrate Exo9's phosphorolytic activity for any eukaryote and we think that this is a significant achievement. This is also the reason why we disagree with the second criticism that "insights into how phosphorolytic activity contributes to the function of the exosome as well as advance over published work are somewhat limited". Besides these two points, we do agree that the specific comments raised by Reviewer 2 address key issues that deserve clarification as detailed below.

Specific points:

1. I am not quite convinced based on the data presented of the functional importance of the phosphorolytic activity. The single RRP41 mutant shows no effect on processing of rRNA (fig 5b). There is some additive effect seen in combination with RRP6, but the phosphorolytic activity seems somewhat redundant with hydrolytic enzymes. Are there RNA targets that are specifically dependent on phosphorolytic activity?

Indeed, we report that the phosphorolytic activity is somewhat redundant with that of the hydrolytic ribonucleases linked to exosome function, especially with the activity of RRP6L2. We think that an overlap between Exo9 and RRP6L2, two distributive ribonucleolytic activities linked to exosome function, is not unexpected. Redundancy is also frequently observed among "independent" RNA degradation pathways. For instance, single mutations in a component of the cytosolic 3'-5' RNA degradation pathway (SKI2) or in a component of the cytosolic 5'-3' pathway (XRN4) have an extremely limited impact on the transcriptome in Arabidopsis. Yet, the corresponding double mutation is lethal (Zhang et al 2005 Science). Also in *E. coli*, and except for the oligoribonuclease gene, single mutations of 3'-5' exoribonucleases are not lethal, whereas several combinations are. All these synthetic lethality reveal redundancy but do not imply that individual enzymes have no functional importance.

The aim and main interest of our study is to demonstrate, using Arabidopsis as a model for plants, that a eukaryotic exosome has an intrinsic phosphorolytic activity. To this end, we show *in vitro* data that demonstrate the phosphorolytic nature of Exo9's activity and we present data that prove that Exo9 is indeed active *in vivo*. Showing that Exo9's activity is involved in the archetypical functions of the exosome during rRNA maturation processes has a significant interest, even if the phosphorolytic and hydrolytic activities overlap to some extent. In fact, and despite that rRNA maturation by-products or 5.8S rRNA precursors are targeted by different activities, we could still identify specific 5' ETS intermediates for Exo9's activity. We do not know yet whether RNA targets that are entirely specific to Exo9's activity exist, but it is entirely possible that they do not. Yet, residues conferring activity to Exo9 are conserved in most plant species (and beyond plants as now discussed in the revised version), suggesting a selective evolutionary pressure to maintain RRP41's activity. This activity might be important to confer robustness rather than specificity to a degradation or maturation process.

Is there any effect on the overall fitness of the plant upon loss of RRP41 activity?

Besides the molecular phenotypes that we initially reported, we can further demonstrate the biological importance of RRP41's activity because we have now identified growth conditions in which the lack of RRP41's activity combined with *mtr4*, *rrp6L2* or *rrp6L2* RRP44 KD mutations results in obvious deleterious effects on plant growth and development. As detailed in our answer to Reviewer 1 above, the double and triple mutants grown in short day conditions do not have a clear macroscopic difference between plants expressing RRP41^{WT} or RRP41^{Pi-Cat-} (see Supplementary Fig. 6a). By contrast, we reproducibly observe a pronounced effect of Exo9 inactivation when plants are grown in long day conditions with LED illumination. While Col-0, *rrp41* RRP41^{WT} or *rrp41* RRP41^{Pi-Cat-} are also in these conditions indistinguishable, the growth of double and triple mutants expressing the inactivated version of RRP41 is strongly impaired as compared to their counterparts with catalytic active Exo9. Pictures that illustrate the plants' phenotypes are presented in the revised version of this manuscript (Fig 6, Supplementary Fig. 6 and 7).

2. Relationship between hydrolytic and phosphorolytic activities should be investigated in more details *in vitro* and *in vivo* in order to understand their contributions.

A detailed investigation of such a relationship *in vitro* requires the reconstitution of the plant exosome from recombinant proteins. Unfortunately, this reconstitution was so far unsuccessful. It was attempted for many months by one of our collaborators. Despite his expertise in this field, this reconstitution failed so far because some of the Arabidopsis Exo9 subunits could not be produced using various expression systems. We explicitly considered this matter in the discussion section: "To investigate such features for the plant exosome □ and to experimentally determine the precise RNA path(s)

leading to RRP41's active site will require the reconstitution of a plant exosome from recombinant proteins, which has failed for technical issues so far. The affinity-purification strategy we employed in this study is not suited to perform biochemical experiments requiring large amounts of proteins. However, its main advantage is that a native, albeit tagged, exosome purified from plants can be studied". Although we are aware that it would have been interesting to provide more biochemical information on the phosphorolytic activity of Exo9, we think that we provide novel data about the existence of a phosphorolytic active eukaryotic Exo9.

Importantly, we also think that we report a quite detailed investigation of the relationship between hydrolytic and phosphorolytic activities using 3'RACE-seq and the P-P1 intermediates as model substrates. This analysis allowed us to identify some intermediates that are specific to Exo9 while other are degraded by redundant activities. In conclusion to this point, our study constitutes the first report of a cooperation between hydrolytic and phosphorolytic activities within a single enzymatic entity.

3. I am not convinced that rRNA processing is affected rather than degradation of the precursor molecule in Fig 5. Is there simultaneous depletion of mature rRNA species (should be included)?

The EtBr staining that was shown below the Northern blot in Fig 5 already indicated similar levels of mature 5.8S rRNAs in all samples. Following the request of Reviewer 2 we re-probed the Northern blots for 5.8S rRNA, and did also not observe a depletion of mature rRNA species (shown in the revised Fig 5). This is probably the result to be expected as we mostly measure mature rRNAs when quantifying RNA samples.

Albeit the steady state levels of mature rRNAs are apparently not reduced, inactivating Exo9 in the *rrp6L2* background results in a phenotype that corroborate a defective ribosome biogenesis in plants, namely a disturbed formation of cotyledon veins. This phenotype, probably caused by a defect in auxin responsiveness, is characteristically observed in ribosomal protein and ribosome biogenesis mutants in Arabidopsis (reviewed in Byrne 2009 and Weis et al. 2015 Trends in Plant Sciences). Cotyledon vein patterns are frequently altered in *mtr4* (Lange et al. 2011 Plant J), but not or rarely in *rrp6L2* single mutants. Importantly, mutating RRP41 in *rrp6L2* mutants does severely impact cotyledon vein formation. About 68% of *rrp6L2 rrp41* RRP41^{Pi-Cat-} cotyledons have disturbed vein patterns, while this problem is only observed in 9% of *rrp6L2 rrp41* RRP41^{WT} cotyledons. This observation mirrors the accumulation of 5.8S precursors and suggests that simultaneous loss of RRP6L2 and Exo9's activity sufficiently affects the efficiency of ribosome biogenesis to alter venation pattern.

We know that neither reduced growth nor aberrant venation patterns are exclusively caused by defective ribosome biogenesis, but as our observation are in full agreement with other reports, (reviewed in Weis et al. 2015 Trends in Plant Science) we think that the mutants phenotypes further support the hypothesis that the phosphorolytic of Exo9s contributes to rRNA processing in Arabidopsis. These new results are now shown in Fig 6, Supplementary Fig. 6 and 7, and described and discussed in the revised text.

We agree that we, as in many studies investigating rRNA processing, can indeed not discriminate with certitude between degradation of the precursor and rRNA processing. Therefore, we usually write 'processing or degradation' as we actually did in the relevant paragraph of the present manuscript "Altogether, these results prove that 5.8S rRNA precursors are endogenous RNA substrates of Exo9's intrinsic activity, and that the exoribonucleolytic activities of RRP44 and RRP6L2 together with Exo9 participate in the processing or degradation of 5.8S rRNA precursors in Arabidopsis." To better reflect this point we have now also modified the subtitle of the paragraph to avoid the term "processing" to "Exo9 contributes to 5.8S rRNA metabolism".

4. For convenience of the readers, Lanes should be labelled in all the figures and referenced in the text . Diagrams describing rRNA processing and the intermediates detected by Northern should be included.

The relevant figures have been modified accordingly, and the lanes are now referenced in the text.

Reviewer #3:

This manuscript reports that the Arabidopsis exosome nine-subunit core (Exo9) has a distributive phosphorolytic activity and participates in the degradation of RNA byproducts and processing rRNA precursors during rRNA maturation. The authors show that Rrp41 in Exo9 is responsible for Exo9's phosphorolytic activity and that Exo9 coordinates with Rrp6L2 and Rrp44 for RNA processing during rRNA maturation. This manuscript thus reports a novel finding that is supported by solid biochemical

data. However, some concerns listed below are required to be addressed.

We thank Reviewer 3 for his/her general comments and we think that we can answer the raised concerns.

1. Rrp41 forms a dimer with Rrp42 in yeast/human exosome, but not Rrp45. It is not clear why the structure model shown in Figure 1c is made for Arabidopsis Rrp41-Rrp45, but not for Rrp41-Rrp42?

The PH-ring of the archaeal exosome is formed by three Rrp41-Rrp42 dimers. In eukaryotes, the PH-ring is formed by the association of three heterodimers. Each hetero dimer is composed of a Rrp41-like subunit associated to a Rrp42-like subunit. Those three heterodimeric pairings are Rrp41-Rrp45, Rrp46-Rrp43, and Mtr3-Rrp42. The dimer formed by Rrp41 and Rrp45 was first inferred from co-expression and co-purification of recombinant proteins and it was finally demonstrated by solving the crystal structure of both the human and yeast exosome (e.g. Liu et al. 2006 Cell; Makino et al. 2013 Nature).

Figure 1 is also not clear. For example, the Mg²⁺ ion is not shown in the model.

We agree and have added the Mg²⁺ ion in the model presented in Figure 1c. Its position was deduced by superimposing our model with the structure of the *E. coli* PNPase (PDB ID: 3GME) (Nurmohamed et al. 2009 J Mol Biol) containing a divalent metal ion.

The phosphate binding residues are not marked in the sequence alignment (Figure 1a and 1b).

The phosphate binding residues are only present in RRP41 subunits (shown in Figure 1a) and not in RRP45/Rrp42 (shown in Figure 1b). The phosphate binding residues are therefore marked only in Figure 1a by a magenta bar above the alignment.

2. It has been shown that the rice Rrp46 forms a homodimer bearing phosphorolytic activity (RNA 16: 1748-1759, 2010). The authors need to compare the sequence and activity of Arabidopsis Rrp41 with that of rice Rrp46. Is it possible that the rice Rrp46 was actually the homolog of Arabidopsis Rrp41?

We did compare the sequences of Arabidopsis and rice RRP46 (thereafter AtRRP46 and OsRRP46, respectively). According to the lab of Hanna S. Yuan (Taipei, Taiwan) who solved the structure of OsRRP46, two residues K75 and Q76 are involved in nucleic acid binding and mutating these residues abolishes the reported *in vitro* ribonucleolytic activity. However, one of those two residues is not conserved in all plants and in particular is not present in Arabidopsis (Q76 is a V in AtRRP46). More importantly, our data show that mutating RRP41 fully abolishes Exo9's activity in Arabidopsis. Therefore, RRP41 is the only active subunit of Exo9 in Arabidopsis, which is in agreement with the fact that RRP41 is the only subunit of the PH-ring to contain all residues required for RNA binding, Pi and Mg²⁺ coordination. Therefore we are quite confident that RRP46 is not active in the context of Exo9 assembly in Arabidopsis. We also do not think that OsRRP46 is the functional homolog of AtRRP41 because the *bona fide* OsRRP41 has all residues to confer phosphorolytic activity to the rice Exo9.

Does Arabidopsis Rrp41 form a homodimer for RNA degradation (that probably can explain its distributive enzymatic activity)? How do we know that the phosphorolytic activity observed in Figure 2 did not result from Rrp41 alone as the protein samples used in this study were purified only by a myc-antibody against myc-tagged Rrp41?

We think it is unlikely that the observed *in vitro* activity can be due to an RRP41 monomer or dimer. First, because the mass spectrometry data do not reveal an enrichment of RRP41 as compared to other subunits of Exo9 (Supplemental figure 1D). Second, because we detect RRP41 only in the context of Exo9 by gel filtration analyses (Supplemental figure 1C). However, to further address this issue, we affinity purified Exo9 by pulling on another subunit tagged with GFP, RRP4. We confirmed that the purified RRP4-tagged Exo9 has a phosphorolytic activity in all points identical to that of RRP41-tagged Exo9. Together with the fact that we abolish Exo9's activity by point mutations in RRP41, our data confirm that the observed activity is only due to RRP41 integrated into Exo9. Those new data have been added to the revised version of the manuscript (new Supplementary Figure 3).

3. The major function of exosome is mRNA degradation. The authors show that the Arabidopsis Exo9 participates in rRNA maturation, however, this manuscript did not answer or discuss the most significant question, if the Arabidopsis Exo9 is involved in mRNA degradation.

The potential involvement of Exo9's activity in degrading mRNAs is worth to investigate and we comment this possibility in the discussion of the revised version. In fact we also looked at a possible effect on mRNAs when we wanted to identify targets to prove that Exo9 is active *in vivo*. RNA seq

data for single activity mutants have been obtained by one of our collaborators but did not reveal a significant deregulation of mRNAs. This result was not completely unexpected because those plants do not have a visible phenotype. The lack of mRNA deregulation might be due to several reasons. Firstly, bulk mRNAs are degraded by redundant cytosolic pathways and we would need several months to produce the appropriate genetic material to investigate this hypothesis, provided that the multiple mutants are viable. Secondly, our current results indicate that Exo9 activity can nibble its substrates (both *in vitro* and *in vivo*) and we have at present no evidence that Exo9's activity has the capacity to degrade long RNA substrates like mRNAs *in vivo*. In conclusion, we agree that testing the potential impact of Exo9 on degrading mRNAs in the cytosol is interesting but we feel that it is clearly beyond the objective of this manuscript, which is to prove that a eukaryotic exosome can have an intrinsic activity. Importantly, we think that major functions of the exosome are linked to specific nuclear functions rather than its involvement in mRNA degradation. rRNA maturation is undeniably amongst the essential cellular processes and showing that the phosphorolytic activity of plant Exo9 participates in this process is definitely answering one of the most significant questions linked to plant Exo9 activity.

REVIEWERS' COMMENTS:

Reviewer #1 (Remarks to the Author):

This already excellent manuscript has been further strengthened by a thorough and complete response to the three reviewers comments. The work presents conclusive evidence that the Arabidopsis RNA exosome core is catalytically active. Such activity has long been speculative, but the current manuscript is the first to provide convincing evidence. This activity is in contrast to what has been reported for yeast and humans and provides major new insight into the eukaryotic RNA exosome. This outstanding manuscript should be published without any further delay.

Reviewer #2 (Remarks to the Author):

The manuscript has greatly improved and all the concerns raised by the reviewers have been fully addressed. The study is now suitable for publication in Nature Communications.

Reviewer #3 (Remarks to the Author):

The authors have well answered my questions.